# Non-canonical function of an Hif-1α splice variant contributes to the sustained flight of locusts

Ding Ding[1], Jie Zhang[2], Baozhen Du[2], Xuanzhao Wang[3], Li Hou[1], Siyuan Guo[1], Bing Chen[3]*, Le Kang[1,2,3]*

[1]State Key Laboratory of Integrated Management of Pest Insects and Rodents, Institute of Zoology, Chinese Academy of Sciences, Beijing, China; [2]Beijing Institutes of Life Science, Chinese Academy of Sciences, Beijing, China; [3]School of Life Science, Hebei University, Baoding, China

**Abstract** The hypoxia inducible factor (Hif) pathway is functionally conserved across metazoans in modulating cellular adaptations to hypoxia. However, the functions of this pathway under aerobic physiological conditions are rarely investigated. Here, we show that Hif-1α2, a locust Hif-1α isoform, does not induce canonical hypoxic responses but functions as a specific regulator of locust flight, which is a completely aerobic physiological process. Two Hif-1α splice variants were identified in locusts, a ubiquitously expressed Hif-1α1 and a muscle-predominantly expressed Hif-1α2. Hif-1α1 that induces typical hypoxic responses upon hypoxia exposure remains inactive during flight. By contrast, the expression of Hif-1α2, which lacks C-terminal transactivation domain, is less sensitive to oxygen tension but induced extensively by flying. Hif-1α2 regulates physiological processes involved in glucose metabolism and antioxidation during flight and sustains flight endurance by maintaining redox homeostasis through upregulating the production of a reactive oxygen species (ROS) quencher, DJ-1. Overall, this study reveals a novel Hif-mediated mechanism underlying prolonged aerobic physiological activity.

*For correspondence:
chenbing@hbu.edu.cn (BC);
lkang@ioz.ac.cn (LK)

**Competing interest:** The authors declare that no competing interests exist.

## Editor's evaluation

The hypoxia-inducible factor (Hif) pathway is well known for modulating cellular responses to hypoxia. Combining physiological and molecular biology insights along with evolutionary analysis, this study shows that a unique Hif-1a splice variant remains active in normoxia and scavenges flight-induced ROS to protect the muscle during the prolonged flight in locusts. This work is novel and as a study in a non-model system, is particularly noteworthy for its elegance.

## Introduction

Flight is a key adaptive strategy for many animals but also poses physiological challenges. Animal flight is the most energetically expensive form of locomotion that relies heavily on mitochondrial aerobic activity. The oxygen consumption rates during flight are approximately 30–150 times higher than those at rest (*Armstrong and Mordue, 1985*; *Bartholomew and Casey, 1978*; *Snelling et al., 2012*). However, the high aerobic performance of flight muscles can produce excessive amounts of reactive oxygen species (ROS) and cause oxidative damage to the myocytes (*Fisher-Wellman and Bloomer, 2009*). The flight activity of *Drosophila* is associated with reduced life span and increased levels of oxidative damage, but oxidative-induced muscle fatigue is rarely observed in flying animals during long-distance migration (*Liechti et al., 2013*; *Magwere et al., 2006*). Flight muscles of a

long-distance flying moth use the pentose phosphate pathway (PPP) in a way that appears to reduce oxidative damage caused by flight (*Levin et al., 2017*). However, under flight conditions, especially during prolonged and continuous flight, the molecular signaling by which flying animals minimize oxidative damage in their muscle systems remains unknown.

The hypoxia inducible factor (Hif) pathway is an evolutionarily conserved oxygen sensing pathway that modulates cellular oxygen level and promotes metabolic responses upon hypoxic induction in animals. This pathway controls hundreds of downstream gene expressions and is an important therapeutic target for many hypoxia-associated diseases in human (*Kim and Kaelin, 2004*; *Masoud and Li, 2015*; *Selak et al., 2005*). The key components of this pathway are Hifs, which comprise an oxygen-sensitive α subunit and a stable β subunit. Under hypoxia, stabilized α subunits heterodimerize with β subunits in the nucleus and induce expression of multiple genes (*Wenger et al., 2005*). In skeletal muscles of mammals, acute exercises lead to reduced myocellular oxygen partial pressure ($PO_2$) and thus stabilize Hif-1α protein, which in turn attenuates ROS production by reducing mitochondrial activity and cellular oxygen consumption (*Ameln et al., 2005*; *Mason et al., 2004*). However, flying animals are able to satisfy the high oxygen demand by the myocytes during flight (*Komai, 1998*; *Meir et al., 2019*). In particular, owing to invagination of the tracheoles into the cells, flying insects conduct oxygen directly into flight muscle cells and exhibit the highest aerobic activity among animals (*Harrison and Roberts, 2000*). Therefore, whether or not the Hif pathway is also involved in flight, a constant physical activity that heavily depends on mitochondrial aerobic metabolism, and how the Hif pathway provides protection against oxidative damage and satisfies the extremely high demands of oxygen consumption are unknown.

In mammals, three Hif-α paralogs (Hif-1α, Hif-2α, and Hif-3α) have been identified; they are derived from genome duplication and encoded by three separated genes, namely, *HIF1A*, *EPAS1*, and *HIF3A* (*Kaelin and Ratcliffe, 2008*). Hif-1α is present in all tissues, whereas Hif-2α and Hif-3α are limited to specific cell types (*Bertout et al., 2008*). Hif-1α enhances oxygen delivery and regulates cellular metabolic adaptation to hypoxia, while Hif-2α promotes vascular endothelial growth by erythropoietin (*Iyer et al., 1998*; *Rankin et al., 2007*; *Zhang et al., 2007*). Due to alternative splicing and different promoters, Hif-3α produces a large number of mRNA variants. The most studied Hif-3α truncated isoform is an inhibitor of Hif-1α and Hif-2α (*Maynard et al., 2005*). Invertebrates possess only one Hif-α gene (Hif-1α) (*Loenarz et al., 2011*). But flybase indicates four alternative transcripts in *Drosophila* and there is further evidence for splice variants in the transcriptomes of other insect species. In insects, the Hif pathway participates in a series of hypoxia-associated biological processes, including tracheal development, diapause, and larval growth (*Centanin et al., 2010*; *Chen et al., 2021*; *Valzania et al., 2018*). In Glanville Fritillary butterfly (*Melitaea cinxia*), a genetic variation of *Sdhd*, which is a regulator of Hif-1α, is associated with altered tracheal volume and flight performance; this association suggests the possible involvement of Hif signaling in flight adaptation (*Marden et al., 2013*; *Marden et al., 2021*; *Pekny et al., 2018*).

The migratory locust (*Locusta migratoria*) is a worldwide agricultural pest capable of long-distance flight (*Ma et al., 2012*; *Wang et al., 2014*). Swarming locusts can migrate hundreds of kilometers per day and invade areas covering millions of square kilometers to an extent, with the help of thermal currents and tail winds (*Zhang et al., 2019*). Moreover, migratory locusts can metabolically adapt to hypoxic environments and make migratory flights on the Tibetan Plateau at altitudes over 3700 m (*Ding et al., 2018*; *Zhang et al., 2013*). As a model for animal flight studies, this species has been employed to reveal many aspects and processes pivotal to the understanding of hormonal regulation of energy material mobilization during flight (*Van der Horst and Rodenburg, 2010*).

In this study, two alternatively spliced Hif-1α isoforms, namely, Hif-1α1 and Hif-1α2, were identified in migratory locusts. The stabilization of Hif-1α1 requires hypoxic induction, whereas, owing to the lack of C-terminal transactivation domain (C-TAD), Hif-1α2 is less sensitive to oxygen and remains stable in normoxia. In vivo, Hif-1α1 plays a classical role in mediating cellular responses to hypoxia. However, Hif-1α2 is an essential regulator of prolonged flight in locusts. Mechanistically, Hif-1α2 is upregulated extensively during flight and modulates glucose oxidation and redox homeostasis in flight muscles. Hif-1α2 is able to transactivate the expression of Dj-1, which is an important ROS quencher, and scavenges flight-induced ROS directly. These findings reveal a regulatory mechanism by which hypoxia signaling balances high aerobic metabolism and the risk of overloaded peroxidation under highly aerobic physiological conditions.

## Results

### Identification of Hif-1α isoforms in locusts

The full-length coding sequences (CDs) of *Hif-1α* (3309 bp) were cloned from locusts. Similar to any other invertebrates, there is only one *Hif-α* paralog in locusts (*Figure 1A*). The predicted protein of locust Hif-1α contained basic helix–loop–helix-PER-ARNT-SIM (bHLH-PAS) domains and two transactivation domains (N-terminal TAD and C-terminal TAD). Sequence alignment analysis revealed that the locust Hif-1α protein shared 51.79%, 51.83%, and 48.3% identity with human Hif-1α (AAC68568.1), Hif-2α (Q99814.3), and Hif-3α (AAH80551.1), respectively. In addition, the locust Hif-1α was conserved at proline and asparagine residues, which are the targets of prolyl hydroxylases (PHDs) and factor-inhibiting Hif (FIH), respectively (*Figure 1—figure supplement 1*). On the basis of full-length transcriptome, two transcripts of *Hif-1α* were observed. The full and short length were named as *Hif-1α1* and *Hif-1α2*, respectively. These transcripts were both derived from the alternative 3′ splice site coupled to the alternative polyA site (*Figure 1B*), and further determined by 3′ RACE (GeneBank access numbers MW349109 for *Hif-1α1* and MW349110 for *Hif-1α2*). The CDs of *Hif-1α2* were 201 bp shorter than those of *Hif-1α1* and the encoded protein lacked the C-TAD, which is required for the transitivity of Hif-1α (*Figure 1C and D*). Evolutionary analysis revealed that such Hif-1α splice form also exists in other Orthoptera (Accession no. ON137898 and ON137899 for *Deracantha onos*), some birds (e.g., XP_025006307.1 for *Gallus gallus* and XP_013038471.1 for *Anser cygnoides domesticus*), and human (NP_851397.1). Additionally, the TADs of Hif-α have varied distributions among insects. In incomplete metamorphosis insects and beetles, the Hif-α protein possesses two TADs (N-TAD and C-TAD), but in flies and moths the C-TAD and its inhibitor FIH are completely missing at the genomic level. Therefore, C-TAD-lacking Hif-1α transcripts, with distinct origins, seem to commonly exist in different insect taxa (*Figure 1—figure supplement 2*). The protein of overexpressed Hif-1α1 was detectable only under hypoxic conditions (1% $O_2$), while the protein of Hif-1α2 was detectable in normoxia and upregulated significantly in hypoxia (*Figure 1E*). Additionally, the protein of Hif-1α2 was observed in the nucleus of cells (*Figure 1—figure supplement 3*). In adult locusts, *Hif-1α1* mRNA was ubiquitously expressed in all tested tissues, but *Hif-1α2* showed the highest expression in flight muscles (*Figure 1F*). In addition, the mRNA levels of *Hif-1α2* were elevated with the maturation of the flight muscles and reached the highest level at the adult stage 12 days post eclosion, while *Hif-1α1* exhibited stable expression at all developmental stages (*Figure 1G*). These results suggest a possible functional division between Hif-1α1 and Hif-1α2 in locusts.

### Hif-1α1 regulates typical cellular responses to hypoxia

To explore the in vivo functions of Hif-1α1 and Hif-1α2 associated with the hypoxic response, hypoxia treatment combined with RNAi assay was performed. *Hif-1α1* and *Hif-1α2* were knocked down by double-strand RNA (dsRNA) injection (*Figure 2A*), and the locusts were subjected to short-term hypoxia (3.5 kPa $PO_2$ for 2 hr). Compared with those of the normoxic control, the hemolymph lactate levels were elevated significantly under hypoxia, and this elevation was attenuated by the knockdown of *Hif-1α1*. Meanwhile, the activities of pyruvate dehydrogenase (PDH), which are negatively regulated by Hif pathway, were significantly repressed under hypoxia and were normalized by the knockdown of *Hif-1α1*. By contrast, the knockdown of *Hif-1α2* had no effects on hypoxia-induced generation of lactate and repression of PDH activity (*Figure 2B*). Additionally, the flight muscle mitochondrial DNA copy number in the ds*Hif-1α1*-injected groups, but not in the ds*Hif-1α2*-injected groups, was reduced significantly under prolonged hypoxia (3.5 kPa $PO_2$ for 13 hr) compared with that in ds*GFP*-injected groups (*Figure 2C*). Thus, the knockdown of locust *Hif-1α1* undermined hypoxic responses and produced cellular damages under hypoxia. Further examination was conducted on the regulatory roles of Hif-1α on PDK and BNIP3, which account for the Hif-mediated PDH activity reduction and mitochondrial biogenesis inhibition. Knockdown of Hif-1α1 and Hif-1α2 had no effect on the mRNA expression of *PDK* and *BNIP3* under normoxic condition (*Figure 2—figure supplement 1A*). But compared with those of the normoxia control, the expression levels of *PDK* and *BNIP3* were upregulated significantly in response to hypoxic induction, and these hypoxia-induced upregulations were repressed by the knockdown of *Hif-1α1* (*Figure 2—figure supplement 1B*). For further confirmation, the Hif pathway was activated in normoxia by silencing PHD, an inhibitor of Hif-α (*Figure 2—figure supplement 1C*). The mRNA expression levels of *PDK* and *BNIP3* were significantly elevated with the knockdown of *PHD*. When *Hif-1α1* and *Hif-1α2* were further knocked down in PHD silenced locusts,

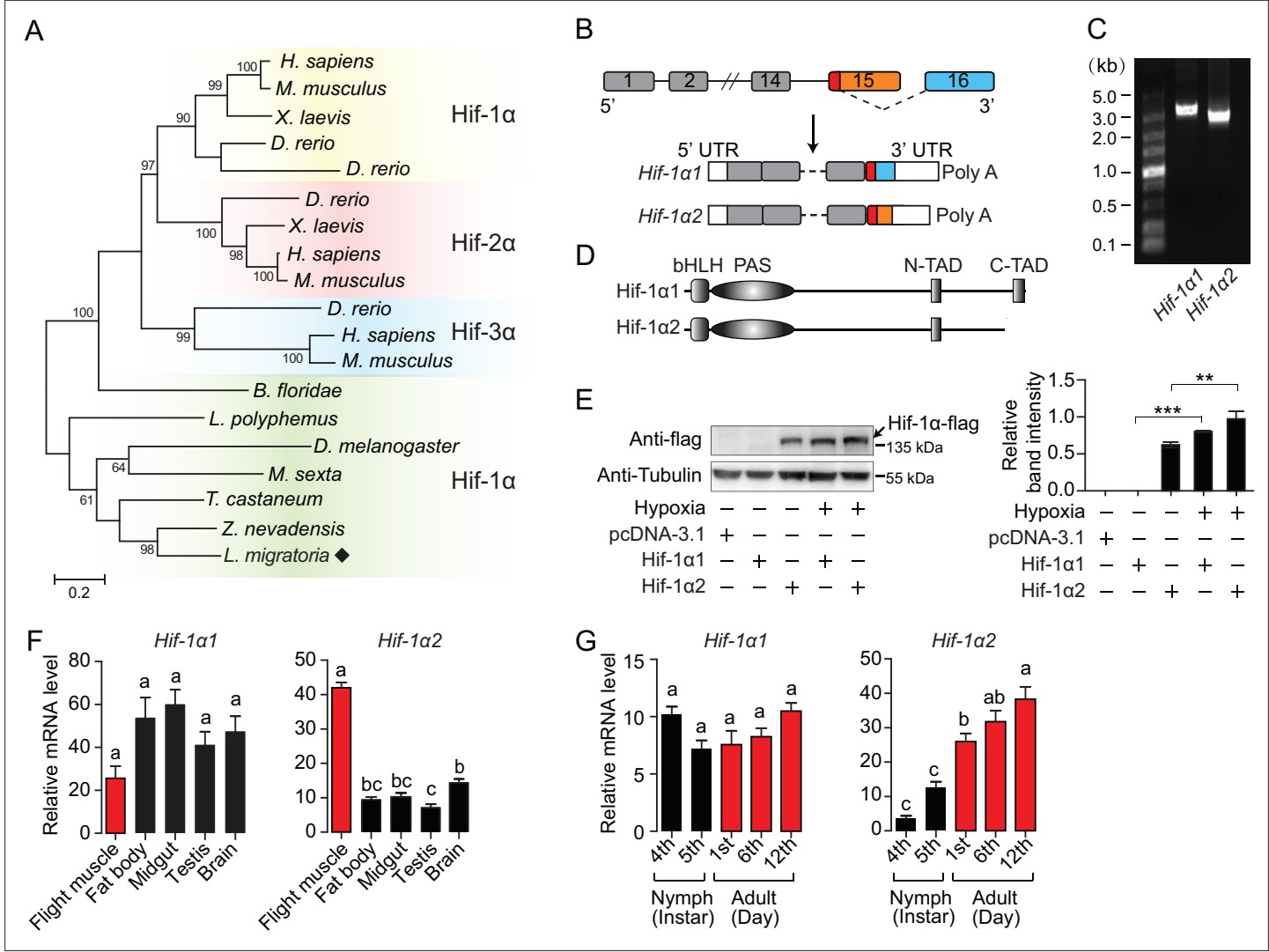

**Figure 1.** An oxygen-insensitive Hif-1α isoform is identified in locusts. (**A**) Phylogenetic tree construction of Hif-α. Amino acid sequences of Hif-α family genes from *Homo sapiens*, *Mus musculus*, *Xenopus laevis*, *Danio rerio*, *Branchiostoma floridae*, *Limulus polyphemus*, *Zootermopsis nevadensis*, *Tribolium castaneum*, *Manduca sexta*, *Drosophila melanogaster*, and *Locusta migratoria* were used for analysis. One thousand bootstraps were performed. Only bootstrap values of over 60 are shown. (**B**) Schematic of *Hif-1α* and its two alternatively spliced isoform mRNAs, *Hif-1α1* and *Hif-1α2*. (**C**) *Hif-1α1* and *Hif-1α2* mRNA expression analyzed by RT-PCR. (**D**) Schematic of Hif-1α1 and Hif-1α2 proteins. (**E**) In vitro expression of Hif-1α1 and Hif-1α2 under different oxygen concentrations. As shown in the Western blotting, Hif-1α2 protein is detectable in normoxia and hypoxia (1% O₂), while Hif-1α1 is detectable only in hypoxia (n=3 replicates). (**F**) Tissue distributions of *Hif-1α1* and *Hif-1α2* (n=3 replicates, 5 locusts/replicate). (**G**) Expression profiles of *Hif-1α1* and *Hif-1α2* in thoracic muscle during development (fourth- and fifth-instar nymphs and first-, sixth-, and 12th-day adults; n=4 replicates, 5 locusts/replicate). The values of the columns are shown as mean ± standard error (s.e.m.). One-way ANOVA with Bonfferoni's test for multiple comparisons, significant differences are denoted by different letters or **p<0.01 and ***p<0.001.

The online version of this article includes the following source data and figure supplement(s) for figure 1:

**Source data 1.** Raw data for spatiotemporal expression profiles of *Hif-1α1* and *Hif-1α2*.

**Source data 2.** Raw data for mRNA and protein expression levels of Hif-1α1 and Hif-1α2.

**Figure supplement 1.** Amino acid alignment of Hif-α in different species.

**Figure supplement 2.** Phylogenetic analysis of Hif-α splice variants across metazoans.

**Figure supplement 3.** Western blot and immunofluorescence analysis showing the protein of overexpressed Hif-1α2 is located in the nucleus of HEK293T cell in normoxia.

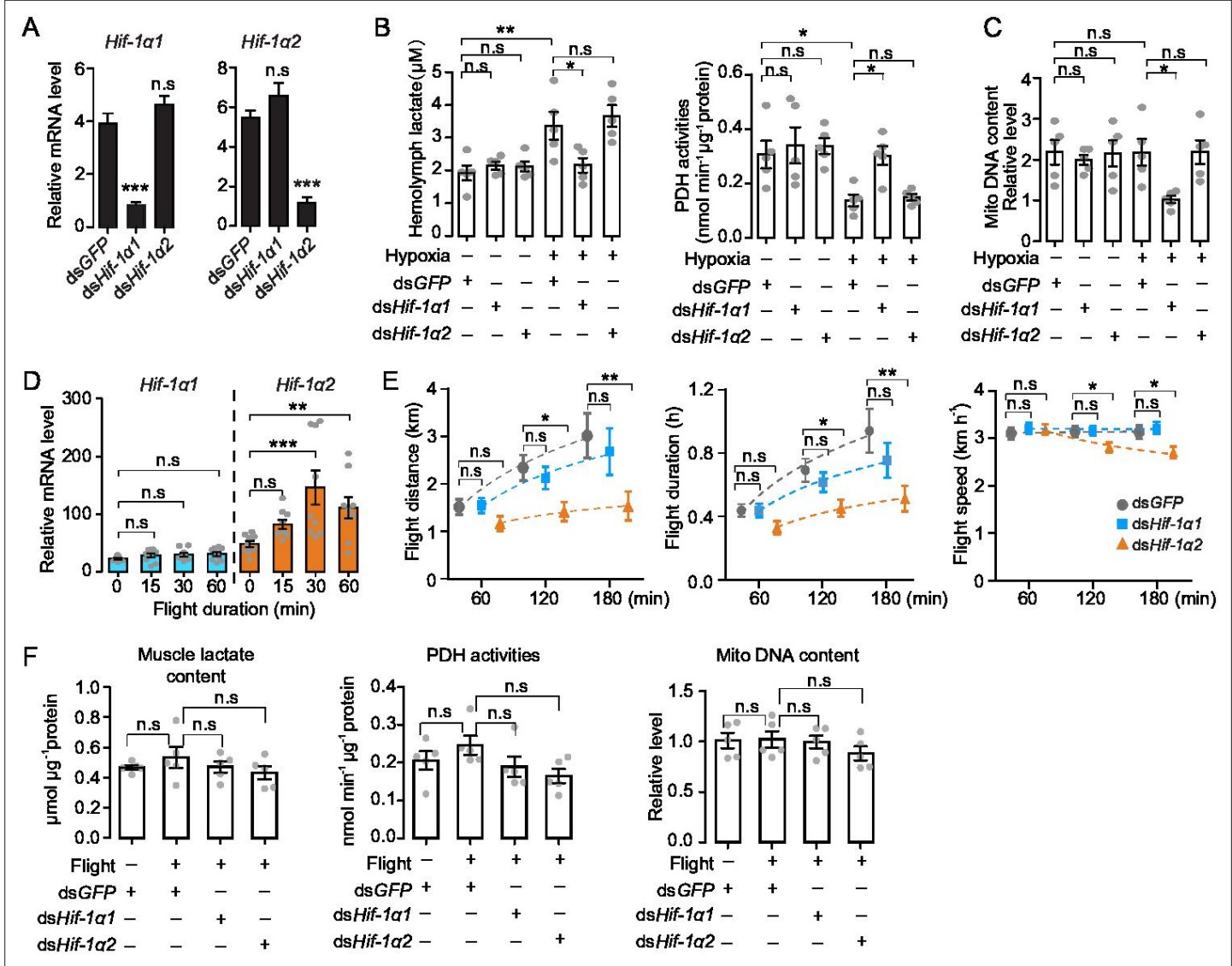

**Figure 2.** Hif-1α1 regulates systematic hypoxic responses, Hif-1α2 is critical for prolonged flight. (**A**) Knockdown of *Hif-1α1* and *Hif-1α2* via injections of ds*Hif-1α1* and ds*Hif-1α2*, with ds*GFP*-injected groups as control (n=5 replicates, 3 locusts/replicate). (**B**) Hemolymph lactate production and pyruvate dehydrogenase (PDH) activity tests upon short-term hypoxic (3.5 kPa *PO₂* for 2 hr) exposure (n=5 replicates, 3 locusts/replicate). (**C**) Mitochondrial DNA copy number measurements upon long-term hypoxic (3.5 kPa *PO₂* for 13 hr) exposure (n=5 replicates, 3 locusts/replicate). (**D**) Expression levels of *Hif-1α1* and *Hif-1α2* under flight treatment (n=9 replicates, 1 locust/replicate). (**E**) Flight performance tests, data for each time point was obtained independently (n≥34). (**F**) Muscle lactate level, PDH activity, and mitochondrial DNA copy number measurements upon flight treatment (n=5 replicates, 3 locusts/replicate). (**A and E**) Student's *t*-test for pairwise comparisons, with ds*GFP*-injected groups as control. (**B, D, and F**) One-way ANOVA with Bonfferroni's test for multiple comparisons. Values are mean ± s.e.m. Significant differences are denoted by *p<0.05, **p<0.01, and ***p<0.001; n.s. represents no significant difference.

The online version of this article includes the following source data and figure supplement(s) for figure 2:

**Source data 1.** Raw data for mRNA expression profile of *Hif-1α1* and *Hif-1α2* under flight treatment.

**Figure supplement 1.** Hif-1α1 regulates metabolic reprogramming upon hypoxic exposure.

**Figure supplement 1—source data 1.** Raw data for *PDK* and *BNIP3* expression levels under hypoxic treatment.

**Figure supplement 2.** Schematic diagram of a flight mill.

**Figure supplement 3.** The Hif-1α1-mediated hypoxic responses cannot be triggered by flight.

**Figure supplement 3—source data 1.** Raw data for *PDK* and *BNIP3* expression levels under flight treatment.

these upregulations were attenuated by the knockdown of *Hif-1α1* (*Figure 2—figure supplement 1D*). However, the knockdown of *Hif-1α2* did not exhibit obvious effects on *PDK* and *BNIP3* expressions under hypoxic or PHD silencing conditions (*Figure 2—figure supplement 1B and D*). Therefore, Hif-1α1, rather than Hif-1α2, modulates the cellular responses to hypoxia in locusts.

### Hif-1α2 is indispensable for sustained flight of locusts

Hif-1α2 is possibly associated with the flight performance of locusts due to its muscle-specific high expression. Thus, forced flight treatment was conducted on locusts and gene expression profiles were examined. Compared with static control, the transcript levels of *Hif-1α2* were significantly upregulated at 30 min of flight ($p<0.001$) and remained by over twofold at 60 min of flight ($p<0.01$). By contrast, the transcript levels of *Hif-1α1* were not upregulated by flight treatment (*Figure 2D*). Furthermore, the effects of Hif-1α on the flight performance were investigated using computerized flight mills (*Figure 2—figure supplement 2*). The results showed that the knockdown of *Hif-1α2* extensively impaired the prolonged flight performance of locusts. Compared with those of the control (injected with ds*GFP*), the flight distance, flight duration, and average flight speed of the *Hif-1α2*-knockdown locust at 120 and 180 min on flight mills were significantly reduced. By contrast, the knockdown of *Hif-1α1* had no obvious effects on flight performance (*Figure 2E*).

To explore the activities of Hif-1α1-mediated hypoxic responses during flight, the mRNA levels of *PDK* and *BNIP3* were tested. Upon flight treatment, no significant upregulations of *PDK* and *BNIP3* mRNA were observed (*Figure 2—figure supplement 3A*). The knockdown of *Hif-1α1* and *Hif-1α2* during flight had no effects on the mRNA expressions of *PDK* and *BNIP3* (*Figure 2—figure supplement 3B*). Moreover, the muscle lactate content, PDH activities, and mitochondrial copy number were measured. Likewise, no changes in muscle lactate levels, PDH activities, and mitochondrial copy number in response to flight or to the knockdown of *Hif-1α1* and *Hif-1α2* were observed (*Figure 2F*). Thus, Hif-1α2 is indispensable for sustained flight of locusts, and flight does not trigger typical hypoxic responses, which are modulated by Hif-1α1 in flight muscles.

### Hif-1α2 regulates glucose metabolic and antioxidative gene expressions in locusts

To determine the regulatory mechanisms of Hif-1α2 in flight performance, we conducted transcriptomic analysis. The gene expression levels of the *Hif-1α1* and *Hif-1α2* knockdown groups were compared with those of the ds*GFP*-injected group. Under normoxic conditions, the knockdown of *Hif-1α1* had a slight effect on the gene expression patterns in flight muscles. No genes had false discovery rate (FDR) lower than 0.05. In the *Hif-1α2* knockdown group, 12 downregulated genes (*MIOX, GAPDH, PGX, GBE, PGI, ENO, DJ-1, PFK, PGM, PYK, LDH*, and *PDHX*) were identified as the candidate target genes (FDR <0.05 and fold change ≥2; *Figure 3A* and *Figure 3—figure supplement 1*). The expression patterns of these genes, except *MIOX*, were confirmed by quantitative real-time PCR (qRT-PCR, *Figure 3B*). Then, *PHD* was silenced to activate the Hif pathway. Meanwhile, *Hif-1α2* was knocked down and the expression patterns of the candidate target genes were tested. Except *MIOX*, all genes were significantly upregulated in the *PHD* knockdown group, and these upregulations were extensively attenuated by the knockdown of *Hif-1α2* (*Figure 3B*). Herein, the 11 target genes of Hif-1α2 were identified. The proteins encoded by *GAPDH, PGX, GBE1, PGI, ENO, PFK, PGM, PYK*, and *LDH* are enzymes involved in glycolysis. The protein encoded by *PDHX* is a core component of the PDH complex, and the protein encoded by *DJ-1* is involved in antioxidation (*Figure 3C*). The regulatory effects of Hif-1α2 on these processes mainly occur in flight muscles and fat bodies but not in the midgut (*Figure 3—figure supplement 2*). Thus, under normoxic conditions, the locust Hif-1α2 may play major roles in regulating glucose metabolism and antioxidative defense.

### Flight-induced elevation of glucose metabolism is Hif-1α2-dependent

To further determine the regulatory effects of Hif-1α on glucose metabolism, the activities of pyruvate kinase (PYK) and phosphofructokinase (PFK) were measured. PYK and PFK are rate-limiting enzymes of glycolysis. Knockdown of *Hif-1α2*, but not *Hif-1α1*, extensively repressed the activities of PYK and PFK (*Figure 4A*). Then, the glycolytic gene expression was examined when the locusts were forced to fly for 60 min after the knockdown of *Hif-1α1* and *Hif-1α2*. The expression levels of *GAPDH, PGX, GBE1, PGI, ENO, PFK, PGM, PYK, LDH*, and *PDHX* were tested via qRT-PCR. The results showed

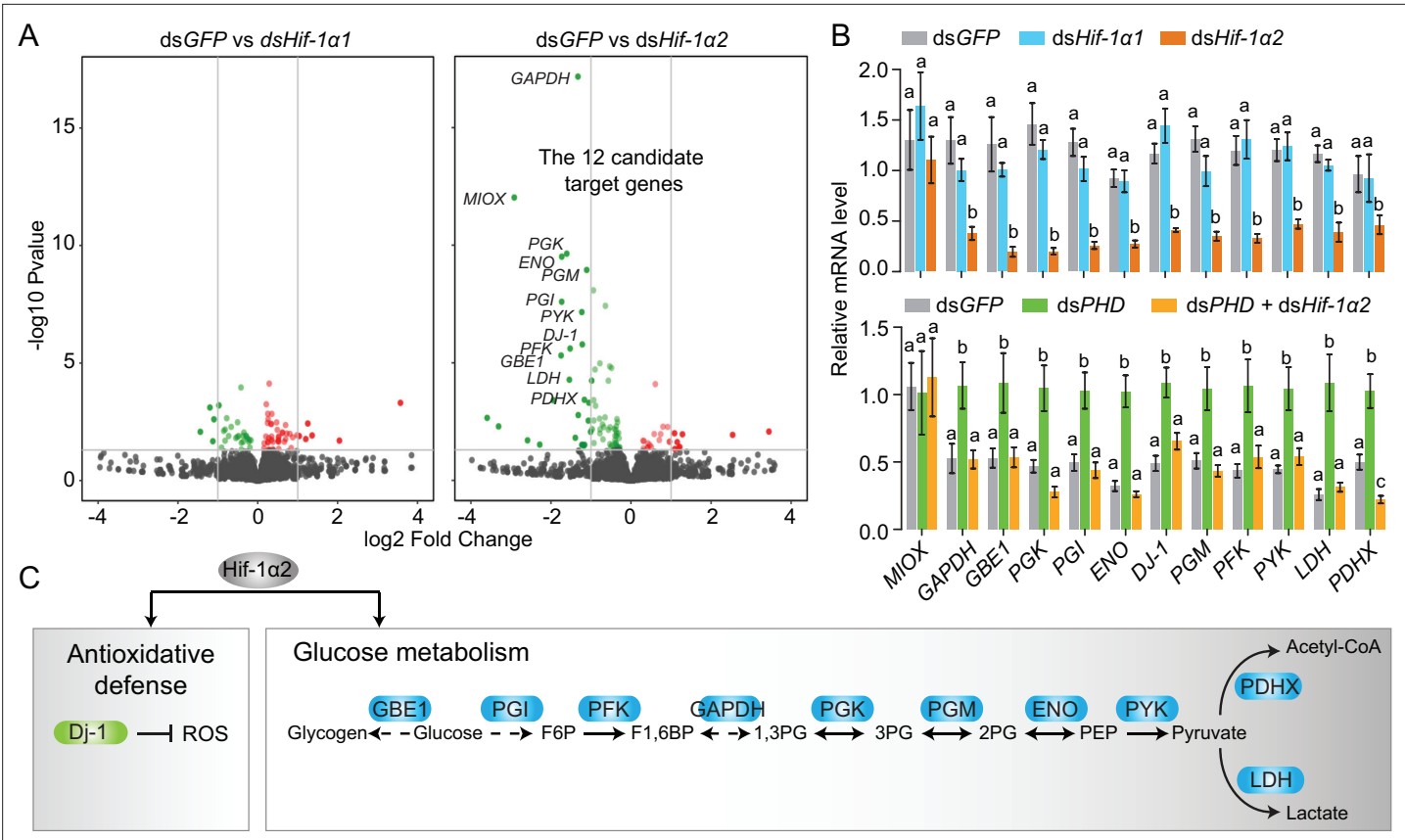

**Figure 3.** Hif-1α2 target gene identification in flight muscle. (**A**) Volcano plot showing a global gene expression pattern of *Hif-1α1* and *Hif-1α2* knockdown groups compared with that of ds*GFP*-injected group in normoxia under resting conditions. Downregulated genes are colored in green, and upregulated genes are colored in red. The top 12 differentially expressed genes with false discovery rate (FDR) values lower than 0.05 are noted by their gene names and identified as candidate target genes of Hif-1α2. (**B**) Quantitative real-time PCR (qRT-PCR) confirmation of the expression patterns of the 12 candidate target genes. Columns are mean ± s.e.m. Significant differences are denoted by different letters (one-way ANOVA with Bonfferoni's test for multiple comparisons, p<0.05, n=4–5 replicates, 5 locusts/replicate). (**C**) Two biological processes regulated by Hif-1α2 under normoxic conditions.

The online version of this article includes the following source data and figure supplement(s) for figure 3:

**Source data 1.** Raw data for quantitative real-time PCR (qRT-PCR) verifications of Hif-1α2 target genes.

**Figure supplement 1.** Heat map showing the expression patterns of candidate target genes of Hif-1α2.

**Figure supplement 2.** Regulatory effects of Hif-1α on its target genes in fat body, brain, and midgut.

**Figure supplement 2—source data 1.** Raw data for quantitative real-time PCR (qRT-PCR) verifications of Hif-1α2 target genes in multiple tissues.

that knockdown of *Hif-1α2*, rather than *Hif-1α1*, dramatically repressed flight-induced upregulation of these genes (***Figure 4B***). In addition, the hemolymph trehalose content reduced significantly in post-flight locusts, but knockdown of *Hif-1α1* and *Hif-1α2* demonstrated no significant effect on hemolymph trehalose level. However, the post-flight locusts exhibited an accumulated muscle glucose content and a reduced pyruvate level in the *Hif-1α2* knockdown group (***Figure 4C***). Thus, the reduction in Hif-1α2 has no obvious effect on hemolymph trehalose uptake but remarkably impairs glucose utilization in flight muscles during flight.

To verify the effect of glucose metabolism on prolonged flight, glycolysis was repressed by silencing *PYK* and the flight performance of the locust was tested. The reduction in *PYK* could repress glycolysis by attenuating the production of pyruvate. However, when the *PYK* in locusts was knocked down, no obvious effects were observed on flight distance, flight duration, and average flight speed in 60, 120, and 180 min tests (***Figure 4D***). Therefore, regulating glucose metabolism may not be the main function of Hif-1α2 in sustaining long-term flight.

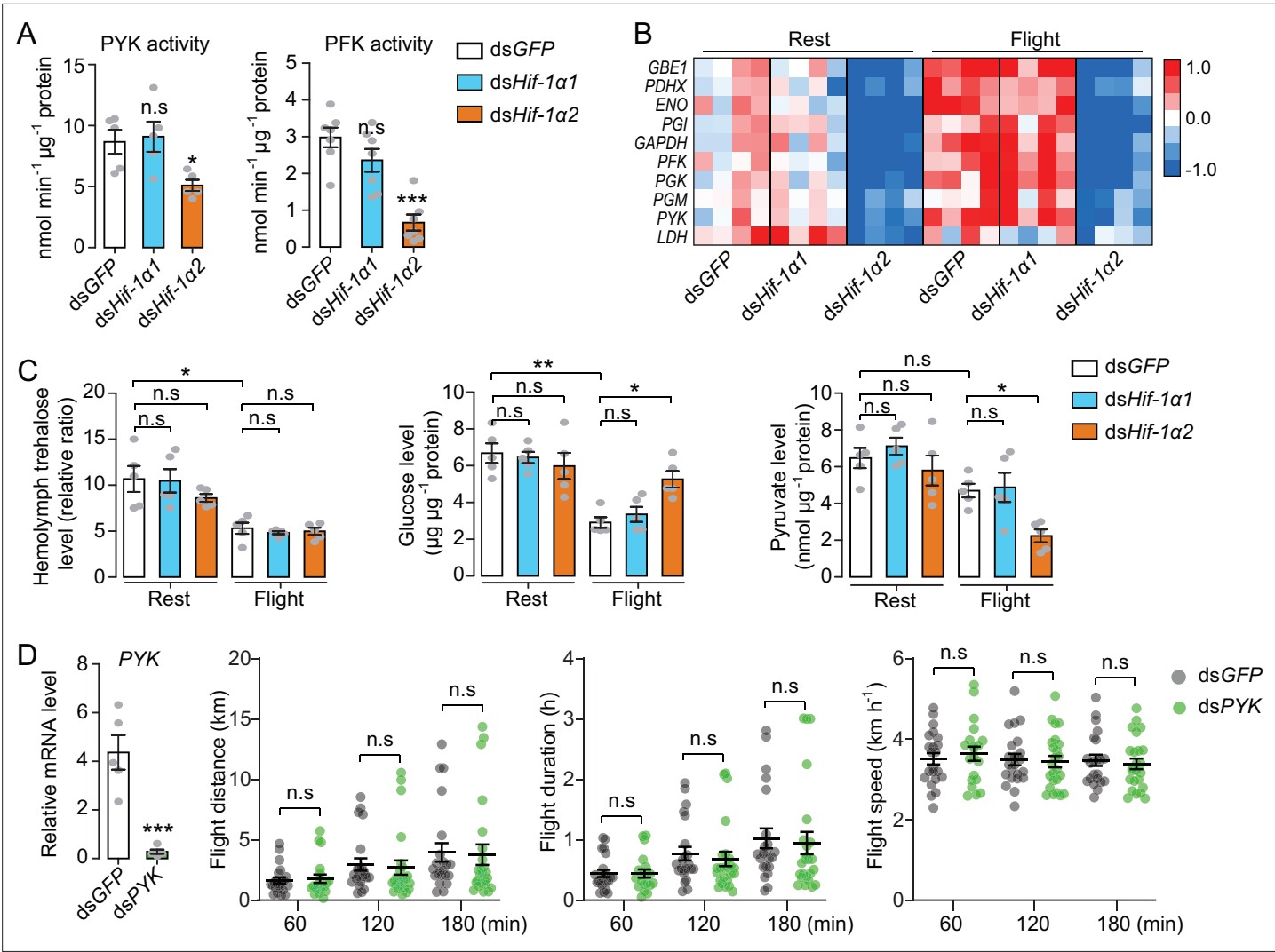

**Figure 4.** Hif-1α2 regulates glucose metabolism during flight. (**A**) Enzyme activities of pyruvate kinase (PYK) and phosphofructokinase (PFK) repressed by knockdown of *Hif-1α2* in flight muscles under resting conditions (n=5–7 replicates, 3 locusts/replicate). (**B**) Flight-induced upregulations of glycolytic genes repressed by knockdown of *Hif-1α2* (n=4 replicates, 3 locusts/replicate). Heat map signal indicates log2 fold change values relative to the mean expression within each group. (**C**) Knockdown of *Hif-1α2* demonstrated no effect on hemolymph trehalose content but inhibited glucose utilization and pyruvate generation in flight muscle during flight (n=5 replicates, 3 locusts/replicate). (**D**) No effect of *PYK* knockdown (n=5 replicates, 3 locusts/replicate) on long-term flight performance (n≥20). (**A and D**) Student's *t*-test for pairwise comparisons, with ds*GFP*-injected groups as control. (**B and C**) One-way ANOVA with Bonfferoni's test for multiple comparisons. Values are mean ± s.e.m. Significant differences are denoted by *p<0.05, **p<0.01, and ***p<0.001; n.s. represents no significant difference.

The online version of this article includes the following source data for figure 4:

**Source data 1.** Raw data for enzyme activities, metabolite contents, and Hif-1α2 target gene expression profiles, and raw data for flight performance tests in the absence of pyruvate kinase (PYK).

## Hif-1α2 plays an antioxidative role during prolonged flight via Dj-1

In addition to glucose metabolism, the target genes of Hif-1α2 are involved in antioxidative defense. We thus knocked down *Hif-1α1* and *Hif-1α2* and tested the effects of flight on oxidative stress. The ROS levels of the flight muscle were measured by dihydroethidium (DHE) staining. At rest, the locusts in the *Hif-1α1* or *Hif-1α2* knockdown and control groups did not exhibit differences in DHE signals. However, in post-flight locusts, the DHE signals in the *Hif-1α2* knockdown group were significantly elevated compared with those in the control. By contrast, the knockdown of *Hif-1α1* displayed no significant differences (*Figure 5A*). After flight treatment, a significant elevation in the $H_2O_2$ content and a reduction in the ratios of reduced to oxidized glutathione (GSH/GSSG) were observed in the

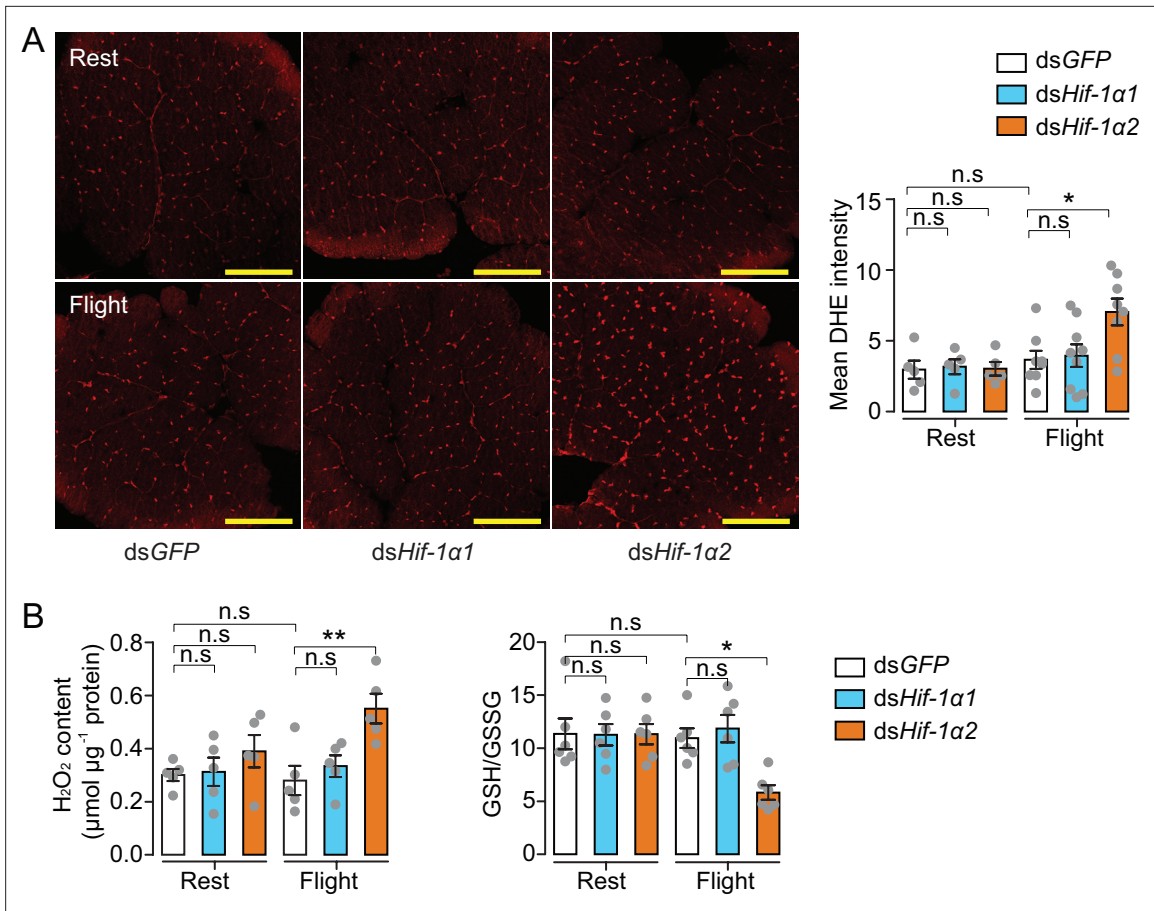

**Figure 5.** Hif-1α2 regulates redox homeostasis during flight. (**A**) High levels of reactive oxygen species (ROS) in flight muscle in the knockdown of *Hif-1α2* locusts during flight. Scale bars: 200 µm. Dihydroethidium (DHE) staining intensity was quantified using ImageJ software (https://imagej.nih.gov/ij, n=5–8 replicates). (**B**) Promotion of $H_2O_2$ production and reduction in the ratios of reduced to oxidized glutathione (GSH/GSSG) by knockdown of *Hif-1α2* during flight (n=5–6 replicates, 3 locusts/replicate). One-way ANOVA with Bonfferroni's test for multiple comparisons. Values are mean ± s.e.m. Significant differences are denoted by *p<0.05 and **p<0.01; n.s. represents no significant difference.

The online version of this article includes the following source data for figure 5:

**Source data 1.** Raw data for dihydroethidium (DHE) staining intensity, $H_2O_2$ content, and reduced to oxidized glutathione (GSH/GSSG) level.

flight muscles of the *Hif-1α2* knockdown locusts, but not in the *Hif-1α1* knockdown locusts (***Figure 5B***). Therefore, in the flight muscle of locusts, the reduction in Hif-1α2 promotes flight-induced ROS generation and impairs the ability to recycle GSSG to its reduced active form (GSH) to control oxidative damage.

Next, the Hif downstream signaling involved in oxidative stress regulation during locust flight was observed. The main focus was *DJ-1*, one of the target genes that was regulated by Hif-1α2 in locusts (***Figure 3C***). The encoded protein of *DJ-1* is an evolutionarily conserved ROS quencher (***Figure 6—figure supplement 1A-C***). In locusts, *DJ-1* displayed the highest expression in the matured flight muscles, it shared a similar expression pattern with that of *Hif-1α2* (***Figure 6—figure supplement 1D***). In addition, the flight-induced upregulations of *DJ-1* were extensively repressed by the knockdown of *Hif-1α2* (***Figure 6A***). At the promoter region of *DJ-1*, three hypoxia response elements (HREs, 5'-RCGTG-3') were identified using the MatInspector software (***Cartharius et al., 2005***). In vitro luciferase reporter assay demonstrated that the overexpression of Hif-1α2 increased the transcriptional activity of the *DJ-1* promoter under normoxia (***Figure 6B***). Electrophoretic mobility shift assays (EMSA) demonstrated that Hif-1α2 could bind to the DNA probe that contained the HRE which was located in the position between −1878 and −1874 bp of DJ-1 promoter (***Figure 6C***). Thus, *DJ-1* could be a direct target of Hif-1α2 in locusts.

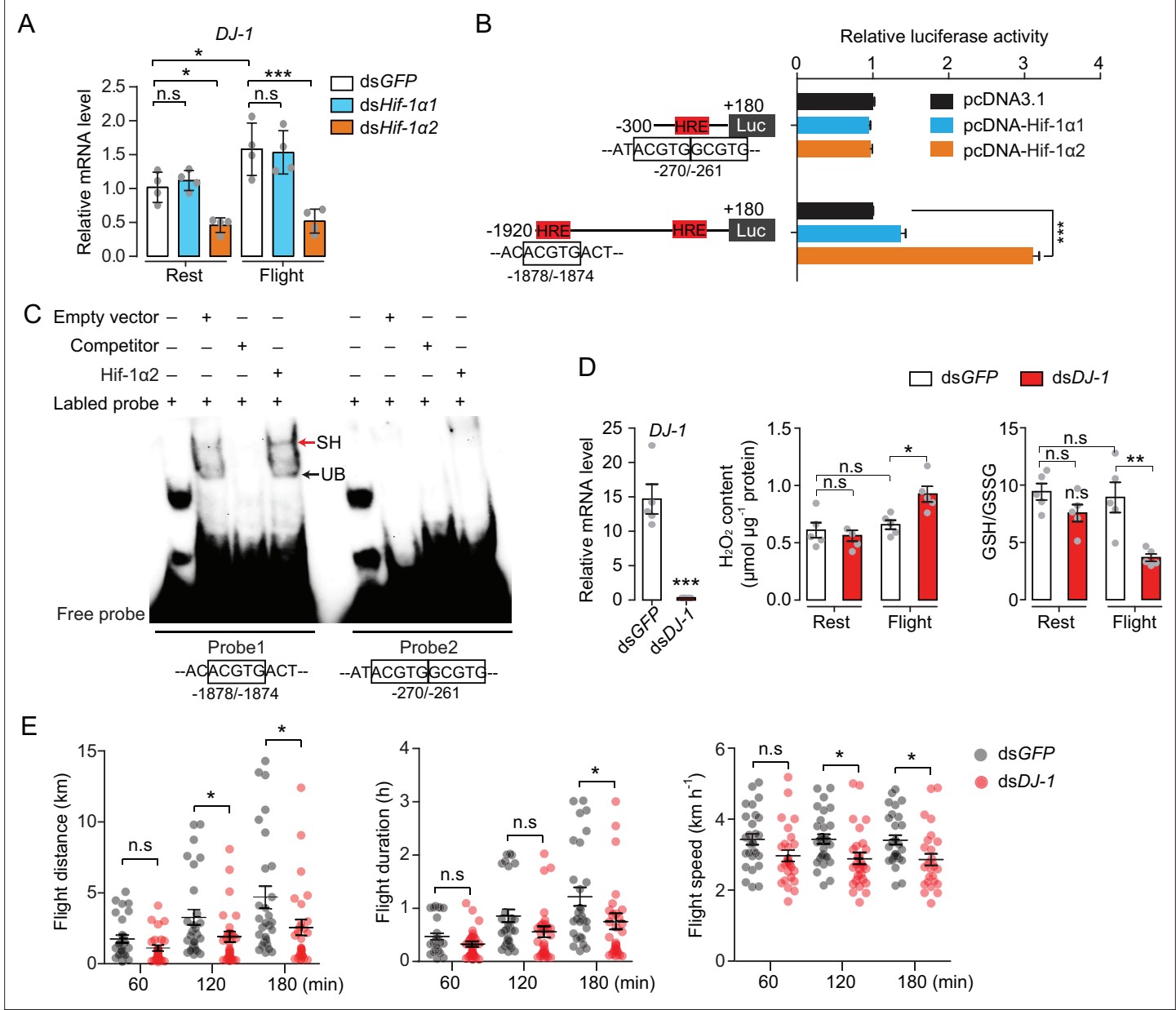

**Figure 6.** Hif-1α2 regulates redox homeostasis via Dj-1. (**A**) Hif-1α2-dependent flight-induced upregulations of *DJ-1* (n=4 replicates, 3 locusts/replicate). (**B**) Dual luciferase assay demonstrating the effect of Hif-1α1 and Hif-1α2 overexpression on *DJ-1* promoter (n=4 replicates). (**C**) Electrophoretic mobility shift assays showing the nuclear extracts which contain Hif-1α2 could bind to the hypoxia response element (HRE) of *DJ-1* promoter. Unlabeled probes were used as competitors. SH represents shift binding of Hif-1α2, UB represents unspecific banding. (**D**) Knockdown of *DJ-1* promoted $H_2O_2$ accumulation, and reduced the ratios of reduced to oxidized glutathione (GSH/GSSG) in muscle during flight (n=5 replicates, 3 locusts/replicate), and (**E**) impaired long-term flight performance of locusts (n≥27). Student's *t*-test for pairwise comparisons. One-way ANOVA with Bonfferroni's test for multiple comparisons. Values are mean ± s.e.m. Significant differences are denoted by *p<0.05, **p<0.01, and ***p<0.001; n.s. represents no significant difference.

The online version of this article includes the following source data and figure supplement(s) for figure 6:

**Source data 1.** Raw data for DJ-1 expression profiles and luciferase assay.

**Source data 2.** Raw data for electrophoretic mobility shift assays.

**Figure supplement 1.** DJ-1 is evolutionarily conserved in animals.

**Figure supplement 1—source data 1.** Raw data for spatiotemporal expression profiles of *DJ-1*.

**Figure supplement 2.** Working model showing the upstream and downstream regulatory details of Hif-1α1 and Hif-1α2 during locust flight.

To determine the antioxidative effects of Dj-1 on flight, *DJ-1* was silenced in locusts and flight treatment was again performed. The content of $H_2O_2$ and the ratios of GSH/GSSG were tested. The reduction in Dj-1 significantly enhanced the flight-induced $H_2O_2$ production and reduced the ratio of GSH/GSSG, compared with the control (*Figure 6D*). In line with Hif-1α2, the knockdown of *DJ-1* significantly repressed the flight performance of locusts in terms of flight distance (p=0.0488) and average flight speed (p=0.0142) in 120 min tests; and flight distance (p=0.0348), flight duration (p=0.0448), and average flight speed (p=0.0149) in 180 min tests on flight mills (*Figure 6E*). Therefore, Hif-1α2 mediates redox homeostasis during locust flight by upregulating the *DJ-1* expression in flight muscle facilitating the maintenance of prolonged flight performance.

## Discussion

The regulatory effect of the Hif pathway on locust flight, which is a highly aerobic physiological process, is determined. Hif-1α2, a newly identified Hif-1α isoform, was stably expressed in flight muscles of locusts regardless of oxygen tension. Hif-1α2 controlled the flight performance by facilitating glucose oxidation and providing redox homeostasis. However, the other isoform Hif-1α1 played a conserved role in response to induction of hypoxia but was not activated by flight treatment (*Figure 6—figure supplement 2*). In general, the activity of Hif-α is strictly controlled by oxygen-tension and the aberrant activation of Hif pathway under normoxia is usually disease-associated (*Koyasu et al., 2018*). But in some specific cell types or physiological conditions, Hif pathway remains active in normoxia. In *Drosophila* crystal cells, the Hif-α protein is stably expressed in normoxia and acts as an activator of Notch signaling to promote the survival of hemocyte during hematopoietic development and hypoxic stress. In these crystal cells, Hif-α is stabilized by NO and does not interact with Hif-β (*Mukherjee et al., 2011*). Vertebrate myeloid cells have shown a similar upregulation of Hif-1a protein in well-oxygenated environments. But myeloid cells have few mitochondria and rely on glycolysis to produce ATP under all conditions (*Cramer et al., 2003*). In this study, the stabilization of locust Hif-1α2 in normoxia was possibly due to the alternatively spliced C-TAD. Additionally, unlike vertebrate myeloid cells, the metabolic profile of flight muscle, which Hif-1α2 was highly expressed, is strictly aerobic. Therefore, the finding of this study greatly extends the functional scope of Hif pathway.

There is an obvious functional differentiation between Hif-1α1 and Hif-1α2 in locusts. Under normoxia, Hif-1α2 displayed a considerably higher expression than the full-length Hif-1α1. Although the protein level and activity of Hif-1α2 were induced by hypoxic treatment or *PHD* knockdown, Hif-1α2 failed to upregulate the gene expression of *PDK* and *BNIP3* and induce hypoxic responses under hypoxia. Therefore, Hif-1α2 had different oxygen sensitivity, being less inhibited by normoxia, and increasing to high levels in low oxygen. Additionally, Hif-1α2 regulated target genes different from those of Hif-1α1. Structurally, Hif-1α1 contains two TADs (N-TAD and C-TAD), but Hif-1α2 only possesses an N-TAD. The Hif-1α splice form without C-TAD was also reported in human as Hif-1α[736]. In line with the locust Hif-1α2, the protein of Hif-1α[736] was detectable in normoxia and was upregulated by the knockdown of PHD2. Meanwhile, Hif-1α[736] could transactivate the promoter of *VEGF* under normoxia (*Berra et al., 2003*; *Gothié et al., 2000*). The functions of TADs in Hif-α are regulating transcriptional activity (*Li et al., 1996*). Each TAD can act independently and is regulated by different mechanisms. An oxygen-dependent degradation domain overlaps with N-TAD and mediates the PHD-dependent degradation of Hif-α (*Bruick and McKnight, 2001*). The transactivation of C-TAD is regulated by FIH-1, a 2-oxoglutarate-dependent oxygenase (similar to PHDs). FIH-1 catalyzes the hydroxylation of an asparagine residue in the C-TAD in normoxia, thereby preventing interaction of Hif-α with the p300/CBP co-activator (*Mahon et al., 2001*). A previous study has shown that, the C-TAD and N-TAD can regulate distinct Hif-dependent gene expression along the oxygen gradient (*Dayan et al., 2006*). Thus, we speculate that due to the lack of C-TAD, locust Hif-1α2 exhibits a reduced oxygen sensitivity and an impaired ability in transactivating gene expressions of *PDK* and *BNIP3*, which are regulated by Hif-1α1.

Hif-1α2 is a transcriptional regulator that modulates glucose metabolism in the flight muscle of locusts. Flight is one of the most energetically expensive forms of locomotion, and carbohydrates and fats are the main fuel types used by animals for flying (*Wegener, 1996*). Locusts use carbohydrates to initiate flight and shift to lipid oxidation for prolonged migratory flight (*Pflüger and Duch, 2011*). Studies on the metabolic regulations of flight mainly focused on neural and hormonal levels thus far. In migratory locusts, octopamine and adipokinetic hormones (AKHs) take part in modulating energy

material mobilization during aggregation and flight (*Ayali and Pener, 1992*; *Ma et al., 2015*). Octopamine stimulates the oxidation of carbohydrates at the initial stage of flight by increasing the content of fructose 2,6-bisphosphate (*Wegener, 1996*). AKHs, which are generated from the corpora cardiac, modulate fat body lipid mobilization through $Ca^{2+}$ signaling, and finally activate triacylglycerol lipase, which catalyze triacylglycerol into the transport form 1,2-diacylglycerol during prolonged flight (*Van der Horst, 2003*). A recent study reported that the AKH/corazonin-related peptide can facilitate muscle lipid utilization through promoting fatty-acid-binding protein production in flight muscles of locusts (*Hou et al., 2021*). In the current study, the reduction in Hif-1α2 extensively repressed the gene expression levels of glycolytic enzymes even in normoxia. Moreover, the flight-induced upregulation of glycolysis was dependent on Hif-1α2. Therefore, Hif-1α2 is possibly located downstream of octopamine or AKH and works as a direct regulator of carbohydrate metabolism during flight. But as mentioned above, utilization of carbohydrate mainly takes place at the initial stage of flight in locusts. That's why repressing glycolysis hardly produced any negative effect on long-term flight.

A prolonged physical activity requires not only a sufficient energy supplement but also a better maintained redox homeostasis in myocytes (*Leeuwenburgh and Heinecke, 2001*). Intense or prolonged physical activity is normally associated with increased ROS production. If this production is not adequately balanced by antioxidants, oxidative damage to biomolecules occurs (*Fisher-Wellman and Bloomer, 2009*). The overloaded ROS attacks the cell membrane fatty acids and activates a chain reaction of lipid peroxidation, finally leading to severe damage to the cellular bilayers or other lipids, proteins, and nucleic acids (*Gaschler and Stockwell, 2017*). Additionally, oxidative damage caused by flight in insects differs depending on their flight physiology, behavior, and life history. A sustained flight throughout life can cause a higher mortality rate to *Drosophila* (*Magwere et al., 2006*). Flight activity of honeybees directly leads to increased oxidative damage, which in turn detrimentally affects their flight performance and foraging ability (*Margotta et al., 2018*). Insects have evolved a series of adaptive strategies to cope with intermittent and migratory flight-induced oxidative stress. Glanville Fritillary butterflies carrying *Sdhd M* allele are associated with the activated Hif signaling, reduced metabolic rate, and larger tracheal volume in larvae, and these associations contribute to less oxidative injury in flight muscle and better flight performance during intermittent flight in adults (*Marden et al., 2021*; *Pekny et al., 2018*). Nectar feeding hawkmoths use their antioxidant stores during migratory flight and through PPP to produce an antioxidant potential to recover from oxidative damage during rest (*Levin et al., 2017*). While the utilization of PPP was reported to be positively correlated with the activation of Hif pathway (*Sadiku and Walmsley, 2019*; *Tokuda et al., 2019*). Therefore, at the molecular level, the Hif pathway likely plays a central role in regulating redox homeostasis during insect flight.

In this study, Hif-1α2 participated in antioxidative processes by upregulating Dj-1 expression during flight. Dj-1 was highly expressed in the flight muscles, and the reduction in Dj-1 significantly impaired the flight performance of locusts and led to accumulated flight-induced oxidative stress in the myocytes of the flight muscles. Dj-1 is a highly conserved protein involved in the regulation of oxidative stress and detoxification. In human, Dj-1 participates in antioxidation by directly quenching ROS upon oxidative modification of a conserved cysteine residue (Cys-106) or by stabilizing Nrf2, which is a master transcriptional regulator of antioxidants (*Clements et al., 2006*; *Taira et al., 2004*). Dj-1 also prevents the formation of a toxic glycolytic intermediate which can cause damages to metabolites and proteins (*Heremans et al., 2022*). Deletion or point mutation (L166P) of Dj-1 is associated with the onset of the familial Parkinson's disease (*Bonifati et al., 2003*). The Dj-1 mutant fruit fly exhibits phenotypes such as male sterility, shortened life span, and reduced climbing ability (*Hao et al., 2010*).

Hif-1α2 confers locusts with a superior ability in modulating redox homeostasis during a prolonged flight while maintaining efficient aerobic performance. In this study, the locust Hif-1α1 regulated the metabolic reprogramming upon hypoxic induction, including promoting lactate production and repressing mitochondrial aerobic activity. However, no such cellular hypoxia-adaptive responses were detected in the flight muscle during flight treatment on flight mills; instead, the redox homeostasis was maintained by Hif-1α2-regulated Dj-1 production. Muscle-specific inactivation of hypoxic responses during flight is speculated to be an adaptive strategy for endurance flight. In mammals, Hif-1α and Hif-2α could maintain skeletal muscle redox homeostasis by attenuating aerobic performance. Hif-1α represses mitochondrial activities by directly regulating the expression of *PDK1* and *BNIP3*, which are both negative regulators of mitochondrial aerobic metabolism (*Kim et al., 2006*; *Zhang et al., 2008*).

Hif-2α can reprogram glucose metabolism from oxidative form to anaerobic form through the activation of the PPAR-α pathway (*Aragonés et al., 2008*). Therefore, in the skeletal muscle of mammals, the activation of the Hif pathways provides oxidative stress tolerance to the myocytes, but impairs aerobic performance as a side effect. However, flight relies more heavily on aerobic metabolism than running of mammals; in particular, flight metabolism is completely aerobic for insects (*Harrison and Lighton, 1998*; *Harrison and Roberts, 2000*). Thus, maintaining an efficient aerobic activity is important for flying adaptation. Functional differentiation of Hif-1α isoforms provides locusts a fine balance between the production of ROS and activity of aerobic metabolism during flight.

Alternative splicing may be a source of functional innovation for Hif-α in locusts. In this study, we found that Hif-1α in locust species generates two transcripts, that is, the full-length *Hif-1α1* and the short-length *Hif-1α2* that lacks the C-TAD domain. The C-TAD of Hif-α is under strong selective pressure in invertebrates; it first appears in non-bilaterians (*Nematostella vectensis*) and has a varied distribution among invertebrates (*Graham and Presnell, 2017*). This domain and its inhibitor FIH are completely absent at the genomic level in some newly emerged insect species, including wasps (Hymenoptera), true flies (Diptera), moths and butterflies (Lepidoptera), all of which are outstanding flyers (*Graham and Presnell, 2017*). Genetic variations in the Hif pathway can affect the tracheal volume and flight performance of lowland butterfly populations under well-oxygenated environment (*Marden et al., 2013*). This evidence combined with our findings implies that the emergence of C-TAD-lacking Hif-1α transcripts is likely to be a substrate for flight adaptation in some insect species.

Insects were the first animals capable of flying , and their respiratory system is the tracheal system. The tracheal supply to flight muscles is divided into three parts: the primary tracheal supply, the secondary tracheal supply and the tertiary tracheal supply. The tertiary tracheal supply is constituted by terminal tracheoles, which conveying oxygen to the mitochondria (*Wigglesworth and Lee, 1982*). During flight, the tracheal system reaches its maximum functional requirements with little reserve capacity (*Snelling et al., 2017*; *Snelling et al., 2012*). Therefore, the tracheal conductance for oxygen and flight performance of insects are closely associated. In the larval stages, the tracheal system becomes oxygen-sensitive, and Hif pathway controls the growth of tracheal terminal branches toward oxygen-starved areas (*Centanin et al., 2010*; *Henry and Harrison, 2004*). The canonical role of the Hif pathway contributes to the ecological adaptation of Glanville Fritillary butterflies from genetically distinct clades (*Marden et al., 2013*; *Marden et al., 2021*; *Pekny et al., 2018*).

The two Hif-1α splices may coordinate their roles in long-lasting flight tasks. In locusts, the canonical role of Hif pathway is modulated by Hif-1α1, which regulates metabolic reprogramming and possibly controls tracheal growth under low oxygen tension. However, the high oxygen conductance of the tracheal system of the locust flight muscle may keep the intracellular oxygen tension above the low level that triggers Hif-1α1 stability. Meanwhile, the relatively easy task of flying with weight support on a flight mill in the present study may render the role of Hif-1α1 in flight muscle undetectable. Nevertheless, when it comes to highly active tissue, Hif-1α1 may provide protective effects too late to prevent oxidative damage. Instead, Hif-1α2, which is expressed in normoxia and has a graded activity with decreasing oxygen, provides continuously variable expression of antioxidant genes so that protection is in place before the damage occurs. This is different from the way Hif-1α1 is typically activated only at very low oxygen tension. As shown in *Figure 1—figure supplement 2*, the similar transcript form of locust Hif-1α2 also exists in some other insect species and birds. Therefore, the Hif-1α2-mediated protective mechanism is possibly applicable to other flying animals, with the locust in this study as the first glimpse.

The regulatory mechanism underlying the spatiotemporal expression of Hif-1α2 remains elusive. Alternative splicing is one of the main sources of spatiotemporally specific mRNA expression and proteomic diversity in eukaryotes. The diverse expression of alternatively spliced mRNA isoforms is usually attributed to alternative promoters or regulatory splicing factors (*Fu and Ares, 2014*; *Russcher et al., 2007*). Alternative promoters can produce a wide variety of transcripts at transcription initiation sites or even affect the splicing patterns of downstream exons (*Zavolan et al., 2003*). The regulatory splicing factors with cell-type-specific expression can bind specifically to enhancers or silencers of a premature mRNA to promote or repress splicing (*Fu and Ares, 2014*). Therefore, alternative usage of promoters or regulatory splicing factors could contribute to the age- and tissue-specific expression of the locust Hif-1α transcripts (*Figure 6—figure supplement 2*). However, detailed mechanism requires further elucidation.

## Materials and methods

### Locust maintenance

Locusts were maintained in the laboratory at the Institute of Zoology, Chinese Academy of Sciences in Beijing. All locusts were reared in ventilated cages ($50 \times 50 \times 50$ cm³) at a density of approximately 100 individuals per cage and fed with fresh wheat seedlings. The culturing environment was kept constant with a 14 hr light:10 hr dark photo regime at 30°C±2°C. Male adults were used for the experiments.

### Flight treatment

The flight treatment is performed using computerized flight mills which have been applied in many published papers (*Du et al., 2022*; *Hou et al., 2021*). The horizontal arm of the flight mill was made of 1.5 mm plastic stick. The radius of the arm was 12 cm, and each revolution was approximately 75 cm. Similar carousel technique for locust flight measurement was firstly reported by *Krogh and Weis-fogh, 1952*. A fan on the top of each flight mill was used to induce flight with a wind speed at 1.5 m/s. The total flight distance, flight duration, and average flight speed were used to represent the flight abilities of the tested locusts. The total flight distance and flight duration were recorded by a computer and the average flight speed was calculated via dividing distance by duration. For flight ability testing, male locusts were harnessed on flight mills for 60, 120, or 180 min at 12 days after eclosion. Individuals with the initial bout of flight over 50 m were seen as flyers. Most of the locust individuals cannot fly continuously during 120 and 180 min of tests, instead they performed intermittent bouts of flight. Thus, the flight duration showing here is the time accumulation of locusts that spent on flight. Twelve-day-old adults that were harnessed on flight mills with the accumulated flight duration reaching 15, 30, and 60 min were selected to induce *Hif-1a* and its target gene expression. For oxidative stress and metabolite measurements, individuals with the flight duration over 45 min in 60 min test on flight mills were selected. The flight muscle was extracted from the mesothorax of locusts and washed with ice-cold phosphate buffered saline (PBS). Muscle tissue from at least three locusts was pooled to create a sample weighed approximately 50 mg. Hemolymph from three to five locusts was collected and centrifuged for 20 min at 4°C at $1000\times g$, yielding approximately 50 µL of cell-free hemolymph. Samples were collected immediately after the flight treatment and placed in liquid nitrogen for further testing.

### Hypoxic treatment

Hypoxic treatment was performed in a hypoxic chamber (FLYDWC-50; Fenglei Co., Ltd, China) in which the ambient temperature and air flow were in automatic control. Twenty male locusts were placed in a cage ($10 \times 15 \times 15$ cm³) and kept in the chamber, into which air was blown and balanced with pure nitrogen to achieve the required $PO_2$ levels. The locusts were maintained at 3.5 kPa $PO_2$ in the chamber for 2 (short-term hypoxia) or 13 (long-term hypoxia) hr at 30°C±1°C; 3.5 kPa $PO2$ is within the range of the critical $PO_2$ calculated in *Schistocerca americana* (2–5 kPa) and *L. migratoria* (3–4 kPa) (*Greenlee and Harrison, 2004*).

### Cloning

The CDS region of *Hif-1α1* and *Hif-1α2* were obtained using nested PCR. The outer primers for *Hif-1α1* were Hif-1α-F1 and Hif-1α1-R1, and the outer primers for *Hif-1α2* were Hif-1α-F1 and Hif-1α2-R1. The inner primers for *Hif-1α1* were Hif-1α-F2 and Hif-1α1-R2, and those for *Hif-1α2* were Hif-1α-F2 and Hif-1α2-R2. The primer sequences are listed in *Supplementary file 1*. The LA Taq DNA polymerase (RR002A, TaKara) combined with 2× GC buffer (RR02AG, TaKara) were used for amplification. The 3′- RACE assay was performed using the SMARTer RACE 5′/3′ Kit (TaKara, CA94043) following the manufacturer's protocol. The primers are listed in *Supplementary file 1*.

### RNAi

RNAi assay was performed to knock down the *Hif-1α1*, *Hif-1α2*, *PHD*, *PYK*, and *DJ-1* expression levels. dsRNA was prepared using the T7 RiboMAX Express RNAi system (Promega) following the manufacturer's protocol. Three µg/µL dsRNA (4.5 µg) was injected into 2-day-old male adult locusts at the second ventral segment of the abdomen. At 10 days post injection, the gene expression levels were examined by qRT-PCR. For double-gene knockdown assay, the 2-day-old male adults were injected

with ds*Hif-1α1* and ds*Hif-1α2* at first and then injected with ds*PHD* 48 hr later. The gene expression levels were examined via qRT-PCR at 8 days post ds*PHD* injection. The ds*GFP*-injected group was used as a control. The primers for dsRNA synthesis are listed in *Supplementary file 1*.

## RNA extraction and qRT-PCR

Total RNA was extracted from the flight muscle, fat body, midgut, and brain by using a TRIzol reagent (Invitrogen). The relative mRNA expression was quantified with a SYBR Green 1 Master Mix (Roche) and a LightCycler 480 instrument (Roche). The relative expression levels of the specific genes were quantified using the $2^{-\Delta Ct}$ or $2^{-\Delta\Delta Ct}$ method, where ΔCt is the Cp value of *Rp49* subtracted from that of the gene of interest. *Rp49* was considered as endogenous controls for mRNAs. At least three biological replicates were assayed for statistical analysis. The qRT-PCR primers are listed in *Supplementary file 1*.

## RNA sequencing and data processing

The flight muscle of three independent replicates was collected for tissue preparation. Total RNA extraction was performed using a TRIzol reagent (Invitrogen), and cDNA libraries were prepared in accordance with the protocols of Illumina. Raw data were filtered, corrected, and mapped to locust genome sequence via HISAT software. The gene expression levels were measured using the criteria of reads per kb per million mapped reads. Differentially expressed genes (DEGs) were detected using the EdgeR software. The differences between the treatment and control groups were represented by p-values and FDR. The DEGs with significance levels at FDR <0.05 in each comparison were considered as the candidate target genes (*Jiang et al., 2019a*). The fastq files of the transcriptome sequence are available at BioProject PRJNA690129. The full-length transcriptome was obtained from *Jiang et al., 2019b*, with NCBI accession number PRJNA517220. Briefly, the full-length transcripts from testes of locusts were enriched by 5'-Cap capturing assay for library preparation; 200 ng of the RNA libraries were loaded on FLO-MIN106 (R9.4) flowcells and were run on a MinION or a GridION X5 (Oxford Nanopore Technologies).

## Enzyme activity measurement

The PYK activity was measured using the Pyruvate Kinase Activity Colorimetric/Fluorometric Assay Kit (BioVision, K709). Approximately 50 mg of flight muscle tissue was homogenized with 200 µL assay buffer and incubated on ice for 10 min. The samples were then centrifuged for 10 min at 4°C at 14,000× *g*. The supernatant was diluted 100 times with the assay buffer for measurement. The values were normalized against lysate protein levels. The PFK activity was measured using the Phosphofructokinase (PFK) Activity Colorimetric Assay Kit (Sigma-Aldrich, MAK093). Approximately 50 mg of flight muscle tissue was homogenized with 500 µL ice-cold assay buffer and centrifuged for 20 min at 4°C at 14,000× *g*. The supernatant was diluted 50 times with the assay buffer for measurement. The values were normalized against lysate protein levels. The PDH activity was measured using the PDH Activity Colorimetric Assay Kit (BioVision, K697). Nearly 50 mg of flight muscle tissue was homogenized with 400 µL assay buffer and incubated on ice for 10 min. The samples were then centrifuged for 5 min at 4°C at 14,000× *g*. The supernatant was diluted 10 times with the assay buffer for measurement. The values were normalized against lysate protein levels.

## Oxidative stress measurement

The ROS level was measured using DHE staining (*Vaccaro et al., 2020*). Flight muscle tissue was collected and placed in room temperature (RT) PBS. The washed tissue was then embedded in optimal cutting temperature compound and immediately placed on dry ice and sectioned at –20°C. Subsequently, 15 µm sections were mounted on microscope glass slides (Thermo Fisher) and incubated for 30 min at 37°C with 2.5 µM DHE (Biorigin, BN11008,). The samples were washed three times for 10 min each with PBS and incubated for 10 min at RT with 1 µg/mL Hoechst (Thermo Fisher, H3570). The samples were washed three times with PBS again and mounted between glass microscope slides and coverslips. The tissue sections were imaged using an LSM 710 confocal fluorescence microscope (Zeiss) at a 10× magnification. The DHE intensity was measured and quantified on ImageJ. The $H_2O_2$ content was measured using the $H_2O_2$ assay kit (Beyotime, S0038). Approximately 50 mg of flight muscle tissue was homogenized with 300 µL assay buffer and centrifuged for 5 min at 4°C at 10,000×

*g*. The supernatant was then used directly for $H_2O_2$ assay. The values were normalized against lysate protein levels. The ratios of GSH/GSSG were measured using the GSH/GSSG Ratio Detection Assay Kit (Abcam, ab138881). Approximately 50 mg of flight muscle tissue was homogenized with 400 µL assay buffer and centrifuged for 10 min at 4°C at 10,000× *g*. The supernatant was diluted five times with the assay buffer for measurement.

## Metabolite measurement

The hemolymph or flight muscle samples from at least three locusts were used for lactate assay. The hemolymph samples were centrifuged for 20 min at 4°C at 1000× *g* to remove the hemocyte. The supernatant was used in lactate assay by using the Lactate Colorimetric/Fluorometric Assay Kit (BioVision, K607). The flight muscle was freshly homogenized with 400 µL PBS on ice and centrifuged for 10 min at 4°C at 10,000× *g*. The supernatant was deproteinized using the 0.5 mL 10 KDa Spin Column (UFC5010BK, Millipore) for lactate assay. Glucose was measured using the Glucose Colorimetric/Fluorometric Assay Kit (BioVision, K606), and the flight muscle from the three locusts was homogenized with 300 µL of $dH_2O$ and boiled for 10 min to inactivate the enzymes. The samples were centrifuged at 14,000× *g* for 15 min. The supernatant was diluted 10 times with $dH_2O$ for glucose assay. The Pyruvate Colorimetric/Fluorometric Assay Kit (BioVision, K609) was used for pyruvate measurement. The flight muscle was homogenized with 300 µL pyruvate assay buffer and centrifuged at 14,000× *g* for 15 min at 4°C. The supernatant was deproteinized with the 10 KDa Spin Column and diluted 10 times with pyruvate assay buffer for measurement. For the flight muscle samples, the measured values were normalized against lysate protein levels. Trehalose content was measured using Agilent 6890N-5973N. For details, 10 µL of cell free hemolymph was diluted for 20 times with $ddH_2O$ and deproteinized by adding with 600 µL methanol and centrifuged at 14,000× *g* for 15 min at 4°C. The supernatant was evaporated with a vacuum concentrator. The dry supernatant was then dissolved in 150 µL newly prepared methoxyamine hydrochloride and incubated at 25°C for 12 hr and then trimethylsilylated by 150 µL *N*-methyl-*N*-(trimethylsilyl)trifluoroacetamide which containing 1% of trimethylchlorosilane. Sucrose (0.1 mg/mL) was used as an internal standard (**Ding et al., 2018**).

## Mitochondrial copy measurement

Total DNA was extracted from the locust flight muscle. The amount of mitochondrial DNA relative to nuclear DNA was determined via qRT-PCR by using primers (**Supplementary file 1**) for *COX2* (mitochondrial genome) and *GADPH* (nuclear genome).

## Cell culture, transfection, and overexpression

The CDs of Hif-1α1 and Hif-1α2 were cloned into pcDNA3.1 expression vector with Flag-tag on the C-terminal ends of the target genes. The HEK293T cells (RRID: CVCL_0063) were seeded in 2 mL of DMEM (Thermo Fisher) in a six-well plate 1 day before transfection. The Hif-1α1 or Hif-1α2 expression vectors (1 µg/well) were transfected to the HEK293T cells by using Lipofectamine 3000 (Thermo Fisher, L3000015). The transfected cells were cultured for additional 36 hr at 37°C for protein expression tests. For hypoxia treatment, the transfected cells were cultured for 30 hr at normoxia (5% $CO_2$ and 95% air) and additional 6 hr at hypoxia (1% $O_2$, 5% $CO_2$, and 94% $N_2$) before the tests. The primers are listed in **Supplementary file 1**. The mycoplasma contamination of the HEK293T cell line was tested using the MycAway Plus-Color One-Step Mycoplasma Detection Kit (Yeasen Biotechnology, 40612ES60) once a month.

## Western blotting

The total proteins of HEK293T cells were extracted using a TRIzol reagent. Nucleoproteins were extracted by using the Nuclear Extraction Kit (Beyotime, P0027). The proteins were subjected to 8% polyacrylamide gel electrophoresis and transferred to polyvinylidene difluoride membranes (Millipore). The membranes were then blocked in 5% (wt. per vol) BSA at RT for 1 hr, followed by incubation with primary antibody (anti-Flag, 1:5000; anti-tubulin, 1:5000; anti-Histone H3, 1:2000) in 3% (wt. per vol) BSA at RT for 2 hr or at 4°C overnight. Secondary antibody (1:5000; ComWin, CW0234S) was used at RT for 1 hr. The immunological blot was detected using the ECL Western Blot Kit (Thermo Fisher). Band intensity was quantified using Quantity One software.

## Immunofluorescence

HEK293T cells were washed three times with ice-cold PBS and fixed with 4% formaldehyde at RT for 15 min. The fixed cells were permeabilized with 0.1% Triton X-100 for 5 min and blocked with 5% goat serum at RT for 1 hr, followed by incubation with primary antibody (anti-Flag, 1:1000) at 4°C overnight. Alexa Fluor 488-conjugated anti-mouse IgG (1:500, Thermo Fisher, R37120) was used at RT in dark for 1 hr. The cell nuclei were stained with 1 μg/mL Hoechst (Thermo Fisher, H3570). A confocal fluorescence microscope (Zeiss, LSM 710) was used for imaging.

## Luciferase report assay

The HEK293T cells were seeded in 500 μL DMEM (Thermo Fisher) in a 24-well plate 1 day before transfection. The promoter region of DJ-1 was cloned into the pGL4.1 vector. The pGL4.1-derived constructs (500 ng/well) were co-transfected with the pcDNA3.1, Hif-1α1, or Hif-1α2 expression vectors (200 ng/well) to the HEK293T cells. The pRL-TK that contains a Renilla luciferase (Rr-luc) encoding sequence (10 ng/well) was co-transfected with the pGL4.1-derived vectors, and used as an internal control to normalize the transfection efficiency and luciferase activity. The cells transfected with different recombinant vectors were cultured for additional 36 hr at 37°C for transcriptional activity analysis by using the Dual-Glo Luciferase Assay System (Promega) with a luminometer (Promega) in accordance with the manufacturer's instruction. The primers used for vector constructions are listed in *Supplementary file 1*.

## Electrophoretic mobility shift assays

EMSA was performed using the Electrophoretic Mobility Shift Assays kit (Invitrogen E33075) according to the manufacturer's protocols. The locust Hif-1α2 was overexpressed in the HEK293T cells. The nucleoproteins of the transfected cells were extracted at 36 hr post transfection by using the Nuclear Extraction Kit (Beyotime, P0027). The oligonucleotides (GCTTGACCAC<u>ACGTG</u>ACTGTCTATT for probe1 and TCATTCAT<u>ACGTGGCGTG</u>AAATCCA for probe2) were labeled with biotin at the 5′ end and dissolved with annealing buffer. The oligonucleotides were incubated at 95°C for 5 min and annealed to generate the double-stranded probe by natural cooling. The unlabeled probes were used as competitors of labeled probes. Oligonucleotide probes were synthesized by Takara (Shanghai, China). DNA-binding reactions were conducted in a 20 μL mixture containing 15 μg of nucleoproteins, 50 ng of poly(dI-dC), 2.5% glycerol, 0.05% NP-40, 5 mM $MgCl_2$, and 20 fmol biotin-labeled probes. For competition assay, 4 pmol unlabeled probes were added to the binding system. The protein-DNA complexes were separated using a 6% DNA retardation gel (Invitrogen, EC6365BOX) in 0.5% TBE buffer and transferred onto nylon membranes. The transferred membrane was then exposed to UV-light for 5 min in a UV-light crosslinking instrument. The membrane was incubated with a streptavidin-horseradish peroxidase conjugate and was detected using the ECL Western Blot Kit (Thermo Fisher).

## Statistical analysis

For transcriptomic analysis, the resulting p-values were adjusted using the Benjamini and Hochberg's approach for controlling the FDR. For quantitative experiments, all data were statistically analyzed using GraphPad Prism 9.0 software and presented as mean ± s.e.m. Student's *t*-test was used for two-group comparison, and one-way ANOVA followed by Bonfferroni's test was used for multi-group comparisons. Differences were considered as statistical significance at $p < 0.05$. All samples were allocated into experimental groups randomly.

## Acknowledgements

This study was supported by: National Science Foundation of China (Grant Nos. 32000332, 32088102, and 31872304); Young Elite Scientists Sponsorship Program by CAST (Grant No. 2019QNRC001).

## Additional information

### Funding

| Funder | Grant reference number | Author |
|--------|------------------------|--------|
| National Science Foundation of China | 32000332 | Ding Ding |
| National Science Foundation of China | 32088102 | Le Kang |
| National Science Foundation of China | 31872304 | Bing Chen |
| Young Elite Scientists Sponsorship Program by CAST | 2019QNRC001 | Ding Ding |

The funders had no role in study design, data collection and interpretation, or the decision to submit the work for publication.

### Author contributions
Ding Ding, Conceptualization, Resources, Data curation, Software, Formal analysis, Funding acquisition, Validation, Investigation, Visualization, Methodology, Writing – original draft, Writing – review and editing; Jie Zhang, Baozhen Du, Software, Methodology; Xuanzhao Wang, Li Hou, Methodology; Siyuan Guo, Software; Bing Chen, Conceptualization, Validation, Investigation, Methodology, Writing – original draft, Writing – review and editing; Le Kang, Conceptualization, Resources, Supervision, Funding acquisition, Investigation, Writing – original draft, Project administration, Writing – review and editing

### Author ORCIDs
Li Hou  http://orcid.org/0000-0001-6727-7053
Bing Chen  http://orcid.org/0000-0002-1238-3948
Le Kang  http://orcid.org/0000-0003-4262-2329

### Decision letter and Author response
Decision letter https://doi.org/10.7554/eLife.74554.sa1
Author response https://doi.org/10.7554/eLife.74554.sa2

## Additional files

### Supplementary files
• Supplementary file 1. Primers used in this study.
• MDAR checklist

### Data availability
The published reference genome of migratory locust used for mapping is available at LocustBase [http://159.226.67.243/download.htm]. Fastq files of the transcriptome sequence for RNAi assay are available at BioProject PRJNA690129. The GenBank accession numbers for the mRNA sequence of locust Hif-1α1 and Hif-1α2 are MW349109 and MW349110, respectively. The GenBank accession numbers for D. onos Hif-1α transcripts are ON137898 and ON137899.

The following datasets were generated:

| Author(s) | Year | Dataset title | Dataset URL | Database and Identifier |
|-----------|------|---------------|-------------|-------------------------|
| Ding D | 2021 | Locusta migratoria Transcriptome or Gene expression | https://www.ncbi.nlm.nih.gov/bioproject/?term=PRJNA690129 | NCBI BioProject, PRJNA690129 |

*Continued on next page*

*Continued*

| Author(s) | Year | Dataset title | Dataset URL | Database and Identifier |
|-----------|------|---------------|-------------|-------------------------|
| Ding D | 2022 | Locusta migratoria hypoxia inducible factor-alpha 1 mRNA, complete cds | https://www.ncbi.nlm.nih.gov/nuccore/MW349109 | NCBI GenBank, MW349109 |
| Ding D | 2022 | Locusta migratoria hypoxia inducible factor-alpha 2 mRNA, complete cds | https://www.ncbi.nlm.nih.gov/nuccore/MW349110 | NCBI GenBank, MW349110 |
| Ding D | 2022 | Deracantha onos hypoxia inducible factor-alpha isoform 1 mRNA, complete cds | https://www.ncbi.nlm.nih.gov/nuccore/ON137898.1/ | NCBI GenBank, ON137898 |
| Ding D | 2022 | Deracantha onos hypoxia inducible factor-alpha isoform 2 mRNA, complete cds | https://www.ncbi.nlm.nih.gov/nuccore/ON137899.1/ | NCBI GenBank, ON137899 |

The following previously published dataset was used:

| Author(s) | Year | Dataset title | Dataset URL | Database and Identifier |
|-----------|------|---------------|-------------|-------------------------|
| Jiang F, Zhang J, Liu Q, Liu X, Wang H, He J, Kang L | 2019 | Locusta migratoria | https://www.ncbi.nlm.nih.gov/bioproject/PRJNA517220 | NCBI BioProject, PRJNA517220 |

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
