## [Editor Report]

The hypoxia-inducible factor (Hif) pathway is well known for modulating cellular responses to hypoxia. Combining physiological and molecular biology insights along with evolutionary analysis, this study shows that a unique Hif-1a splice variant remains active in normoxia and scavenges flight-induced ROS to protect the muscle during the prolonged flight in locusts. This work is novel and as a study in a non-model system, is particularly noteworthy for its elegance.

---

## [Decision Letter]

**Decision letter after peer review:**

Thank you for submitting your article "Non-canonical Function of a Hif-1α Splice Variant Contributes to the Sustained Flight of Locusts" for consideration by *eLife*. Your article has been reviewed by 3 peer reviewers, and the evaluation has been overseen by a Reviewing Editor and Utpal Banerjee as the Senior Editor. The following individuals involved in review of your submission have agreed to reveal their identity: James Marden (Reviewer #1); Jiwon Shim (Reviewer #3).

Essential revisions:

1. Provide a taxonomic view of the presence of Hif-1a splice variants in locusts and other insects.

2. Include a detailed supplementary figure and text to support Figure 6F showing a hypothesis that summarizes both the upstream and downstream regulatory details would help readers form a mechanistic understanding.

3. State clearly the hypothesis of the role of Hif-1a2 in putting anti-oxidant protection in place before the damage occurs.

4. Strengthen the discussion by considering other findings regarding oxidative damage caused by flight in insects that differ in their flight physiology, behavior and life history.

5. Ensure that the entire paper is corrected for and free of errors in citing the literature.

6. Include an account of statistical analysis performed in the Methods.

*Reviewer #1 (Recommendations for the authors):*

L. 32: Omit "constructing".

L. 36-38: This sentence discusses birds, but one of the two cited papers is about sphingid moths (Levin and Davidowitz 2017) and the other (Skrip et al., 2015) has no detailed physiological analysis and is not informative here. Hence, omit the tangential mention of birds, cite only the Levin paper and say something like "flight muscles of a long-distance flying moth use the pentose phosphate shunt in a way that appears to reduce oxidative damage".

L. 39: Given the Levin paper, which identifies a particular carbon atom within glucose as important for maintaining redox homeostasis during flight, the sentence in L. 39 seems unjustified. The present paper is more about signaling pathways that regulate processes that minimize oxidative damage during flight, so craft this sentence to say something highlighting the general lack of information in that realm.

L. 44-45: The Rytkonen and Storz 2011 paper is about evolutionary origins of oxygen sensing. It's not a primary reference for hypoxia-associated diseases in humans. Better citations would be https://pubmed.ncbi.nlm.nih.gov/15611513/ and https://pubmed.ncbi.nlm.nih.gov/15652751/

L. 44: "an important therapeutic target" The cited Gonzalez et al., 2018 paper discusses the potential of this pathway for therapies, but other than one VEGF treatment cited in that paper, it is not a reference you can use to support this statement.

L. 61-63: "Hif-1α enhances oxygen delivery and regulates cellular metabolic adaptation to hypoxia, while Hif-2α promotes vascular endothelial growth by erythropoietin (Semenza, 1999, Zhao et al., 2015)." These papers do not support this statement. The Semenza paper discusses the role of Hif-1a in regulation of EPO; the Zhao et al., paper has a very nice overview of Hif-2a function but never mentions erythropoietin. At this point I’ve only made it to line 63 and I’ve found multiple instances of the references not aligning with the text. I don’t have time to fact check all of your references, so I’m not going to do that anymore, but the editor should firmly instruct you to go through the entire paper and make sure that you are correctly citing the literature.

L. 66: "Invertebrates have only one Hif-α paralog (Hif-1α).” I think you should finish the thought stimulated by the prior sentence about splice variants in the Hif family genes of vertebrates. This sentence might say “Invertebrates have only one Hif-1α gene, but flybase indicates four alternative transcripts in *Drosophila* and there is further evidence for splice variants in the transcriptomes of other insect species.” You could revisit this in the discussion, and in so doing cover what for me is a weak point of the manuscript: is this splice variant unique to migratory locusts, orthoptera in general, or is it more broadly present in insects but heretofore overlooked? You show that in a supplementary figure but don’t discuss it.

L 69: "associated with altered flight performance; this association suggests the possible involvement of Hif signaling in flight adaptation (Marden et al., 2013).” I think you should cite two followup studies (https://journals.biologists.com/jeb/article/221/6/jeb171009/246/Enzyme-polymorphism-oxygen-and-injury-a-lipidomic; https://pubmed.ncbi.nlm.nih.gov/32895932/) that discuss oxidative damage, larval development, and additional aspects of flight in that species. Doing so will point readers to a broadly relevant set of studies that examine intermittent flight and ischemia reperfusion injury caused by repeatedly flying to fatigue. Making a distinction between migratory vs. intermittent fliers may stimulate important insights as work in this area progresses.

L. 76: "make migratory flights on the Tibetan Plateau”. Please indicate an approximate or range of altitudes where this occurs.

L. 80-90: I like this preview summary of the results, but if editors insist on a length reduction, you could save this for the Results or Conclusion sections and instead say something brief, e.g. “Here we characterize two alternative splice variants of locust Hif-1a and test the hypothesis that they perform different roles in flight physiology and migratory ability.”

L. 101: "On the basis of full-length transcriptome, two transcripts of Hif-1α were observed.” It is not obvious at this point or in the methods how this transcriptome was made (whole body? Flight muscle?) and if/where it is accessible by others.

L. 106: "encoded protein lacked the C-TAD”. Extend this sentence by explaining what this domain does. This will inform the hypothesis that these isoforms have different functions.

L 106-109: You highlight the presence of similar splice variants in vertebrates but don’t mention other insects. Your supplementary figure showing this diversity is very nice and provides an overview of the diversity of Hif-1a splice variation, along with boxes that I think show presence/absence of Hif regulatory genes in the genome (figure legend show define those boxes unambiguously). Spend a sentence or two describing what you see across insects, and perhaps in the discussion, how well you think insects have been carefully surveyed for splice variants, and what your findings may imply for insects other than locusts.

L. 263: Not clear if "In vertebrate" is a typo referring to Invertebrates or is something that occurs "in vertebrates".

L. 274: "Therefore, Hif-1α2 was less sensitive to oxygen tension than Hif-1α1". Figure 1E indicates that Hif1a2 reaches higher levels in hypoxia, so it is arguably more sensitive to low oxygen tension (whereas your point is based on it starting from a higher baseline). To make this easier to understand, just say that“"Hif1a2 had different oxygen sensitivity, being less inhibited by normoxia, and increasing to high levels in low oxygen"

L 335-336: "However, no such cellular hypoxia-adaptive responses were detected in the flight muscle during flight." Somewhere you should introduce the idea that the abundant tracheal system of locust flight muscle, and the relatively easy task of flying with weight support on a flight mill, may have kept the intracellular oxygen tension above the low level that triggers Hif-1a1 stability and canonical activation of hypoxia responsive genes. If you introduce that concept, it recognizes that there may be a role for Hif-1a1 in flight muscle that your experiments did not reveal, and helps form the hypothesis that Hif-1a2 has an earlier and graded activity, from a low level at normoxia and increasing as oxygen decreases, thereby protecting against oxidative damage at all levels of exertion rather than as an emergency response that may lag behind the damage.

L 355-356: "On the other hand, our study provides a potential insecticide target for locust plague control." This is presented as an afterthought. Either omit, or if you think it is an important point, begin a new paragraph and develop the argument a bit more. For instance, do you envision this as an RNAi treatment? What would be the major challenges in targeting and minimizing off-target effects, both within locusts and in other insects that one would not want to damage.

Figure 5A: The oxidized DHE appears to be elevated in spaces between cells or groups of cells (myofibers). To my eye, those look like tracheae, and the red dots within fibers might be cross sections of tracheoles where they penetrate into cells. The authors may be stimulated to consider a coupled role of tracheae and muscle cells in sensing falling O2 concentration and functioning together to protect both cell types against oxidative damage. Due to their thin-walled and membranous nature, it seems possible that tracheae may be more sensitive to oxidative damage than muscle, or that muscle has high basal expression of a number of anti-oxidants but tracheae do not. The authors may wish to discuss such a scenario, and testable hypotheses that may emerge from this notion. Also, my copy has fairly low resolution images of these micrographs. The authors should make sure that their highest resolution images are published.

Figure 6 C: Competitor is misspelled.

Figure 6: Panels B: why are the these Hif genes labeled PC (which could be confused for "positive control")?

Panel C: What are the SH and UB bands in panel C, and where in the figure is a shift evident?

Please provide a bit more information for people like me who aren’t familiar with the details of this type of gel shift assay.

*Reviewer #2 (Recommendations for the authors):*

Although I am not a molecular biologist, I was impressed with the comprehensiveness of the work. My only main source of confusion was the link between Hif-1alpha2, the upregulation of glucose oxidation, and its role in facilitating prolonged flight. As you state later in the paper, locust flight is only initially powered by carbohydrate catabolism, and sustained locust flight is instead powered by lipid catabolism. I could be wrong, but I suspect that upregulation of glucose oxidation may rather help to augment the power output of initial flight, when locust metabolic requirements are greatest as they must power the requirements of ascent. Thereafter, locusts switch to lipid catabolism and more prolonged flight is also facilitated by the use of thermal currents and tail winds.

*Reviewer #3 (Recommendations for the authors):*

1. Despite the relatively high levels of Hif-1a2 in resting flight muscles, the authors have shown that Hif-1a2 is further induced after the long flight (30-60min), which would elevate the levels of Hif-1a2 target genes (Figure 2D). And given that loss of Hif-1a2 does not cause any flight defects in early time points (Figure 2E), it is assumed that an increase in the level of Hif-1a2 upon long flight is critical for its function and target genes. Thus, it would have been more appropriate to perform the transcriptome analysis (Figure 3A) after the long flight to better understand transcriptional targets of elevated Hif-1a2 during the flight. Current data may not reflect complete transcriptomic changes of flight muscles after the Hif-1a2 induction and would indicate targets of Hif-1a2 in resting states. As a minimum, the 12 differentially expressed genes need to be validated before and after the long flight and in the presence and absence of Hif-1a2 to better verify the expression and function of glucose metabolism genes and DJ-1.

2. In the same vein, 0-hour controls for dsHif-1a1 or dsHif-1a2 are missing in Figure 4B, Supplementary Figure 4, and Figure 6A. The authors need to clarify whether Hif-1a2 activates downstream targets in resting conditions compared to the long-term flight.

3. For the flight measurements in Figure 2E, the authors observed flight distance, duration, and speed up to 180 minutes for flight abilities, from which loss of Hif-1a2 showed phenotypes after 120 minutes. In Figure 4D and Figure 6E, the authors also measured the flight performance, but experiments do not seem to be identical to what is shown in Figure 2E, according to methods. The evaluation of flight performance in dsPyk or dsDJ-1 backgrounds needs to be identical to what is shown in Figure 2E as lack of energy would likely impact long flight behaviors. If it is a matter of graph formatting, the same format needs to be presented to directly compare phenotypes of different genotypes.

---

## [Author Response]

Essential revisions:1. Provide a taxonomic view of the presence of Hif-1a splice variants in locusts and other insects.

Thank you, we have added a detailed description in line 108-116 and related discussion on this issue in line 366-377.

2. Include a detailed supplementary figure and text to support Figure 6F showing a hypothesis that summarizes both the upstream and downstream regulatory details would help readers form a mechanistic understanding.

We have added a supplementary figure (Figure 6—figure supplement 2) to show a mechanistic elaboration on the regulatory pathways and also a discussion (line 404-414) on this issue.

3. State clearly the hypothesis of the role of Hif-1a2 in putting anti-oxidant protection in place before the damage occurs.

We have added a paragraph to discuss this issue in line 390-403

4. Strengthen the discussion by considering other findings regarding oxidative damage caused by flight in insects that differ in their flight physiology, behavior and life history.

We have added discussion on this issue in line 321-335.

5. Ensure that the entire paper is corrected for and free of errors in citing the literature.

We have corrected all the errors in literature citation.

6. Include an account of statistical analysis performed in the Methods.

We have added an account of statistical analysis in line 625-631 in the Methods.

Reviewer #1 (Recommendations for the authors):L. 32: Omit "constructing".

Thank you, it has been revised (Line 32)

L. 36-38: This sentence discusses birds, but one of the two cited papers is about sphingid moths (Levin and Davidowitz 2017) and the other (Skrip et al., 2015) has no detailed physiological analysis and is not informative here. Hence, omit the tangential mention of birds, cite only the Levin paper and say something like "flight muscles of a long-distance flying moth use the pentose phosphate shunt in a way that appears to reduce oxidative damage".

Thank you, it has been revised as suggested.

**“**Flight muscles of a long-distance flying moth use the pentose phosphate pathway (PPP) in a way that appears to reduce oxidative damage caused by flight (Levin et al., 2017).” (Line 36)

L. 39: Given the Levin paper, which identifies a particular carbon atom within glucose as important for maintaining redox homeostasis during flight, the sentence in L. 39 seems unjustified. The present paper is more about signaling pathways that regulate processes that minimize oxidative damage during flight, so craft this sentence to say something highlighting the general lack of information in that realm.

Thanks for the suggestions, we have revised as “However, under flight conditions, especially during prolonged and continuous flight, the molecular signalling by which flying animals minimize oxidative damage in their muscle systems remains unknown.” (Line 37)

L. 44-45: The Rytkonen and Storz 2011 paper is about evolutionary origins of oxygen sensing. It's not a primary reference for hypoxia-associated diseases in humans. Better citations would be https://pubmed.ncbi.nlm.nih.gov/15611513/ and https://pubmed.ncbi.nlm.nih.gov/15652751/

Thank you, we have revised as suggested, ie., (Kim and Kaelin, 2004; Selak et al., 2005) (Line 43)

L. 44: "an important therapeutic target" The cited Gonzalez et al., 2018 paper discusses the potential of this pathway for therapies, but other than one VEGF treatment cited in that paper, it is not a reference you can use to support this statement.

Sorry for the mistakes in citation. We have replaced Gonzalez et al., 2018 by Masoud, G.N. and Li, W. 2015. This review summarizes the therapeutic approaches targeting Hif pathway. (Line 44)

Ref: Masoud GN, and Li W. 2015. HIF-1alpha pathway: role, regulation and intervention for cancer therapy. Acta Pharm Sin B *5*: 378-389.

L. 61-63: "Hif-1α enhances oxygen delivery and regulates cellular metabolic adaptation to hypoxia, while Hif-2α promotes vascular endothelial growth by erythropoietin (Semenza, 1999, Zhao et al., 2015)." These papers do not support this statement. The Semenza paper discusses the role of Hif-1a in regulation of EPO; the Zhao et al., paper has a very nice overview of Hif-2a function but never mentions erythropoietin. At this point I've only made it to line 63 and I've found multiple instances of the references not aligning with the text. I don't have time to fact check all of your references, so I'm not going to do that anymore, but the editor should firmly instruct you to go through the entire paper and make sure that you are correctly citing the literature.

We apologize for these incorrect literature citation. We have given a thorough checking of all the references, and made the following citation correction:

1)We replaced the papers Semenza, 1999 and Zhao et al., 2015 by the papers Zhang H et al., 2007, Iyer NV et al., 1998, and Rankin EB et al., 2007(Line 62). Among these, the two papers Zhang H et al., 2007 and Iyer NV et al., 1998 revealed the role of Hif-1α in metabolic reprogramming and oxygen transduction in cancer cells and embryonic stem cells. The paper Rankin E. et al., 2007 provided important evidence that Hif-1α and Hif-2α have distinct roles in the regulation of hypoxia-inducible genes and that EPO is preferentially regulated by Hif-2α in the liver.

2) Additionally, we have ommited Jenni-Eiermann et al., 2014 in Line 35, replaced Lavista-Llanos et al., 2002 by Centanin et al., 2010 in Line 69 and replaced Suarez et al., 1990 by Wegener, 1996 in Line 297.

References:

Zhang L, Lecoq M, Latchininsky A, and Hunter D. 2019. Locust and Grasshopper Management. Annu Rev Entomol *64*: 15-34

Iyer NV, Kotch LE, Agani F, Leung SW, Laughner E, Wenger RH, Gassmann M, Gearhart JD, Lawler AM, Yu AY*, et al.* 1998. Cellular and developmental control of O2 homeostasis by hypoxia-inducible factor 1 α. Genes Dev *12*: 149-162.

Rankin EB, Biju MP, Liu Q, Unger TL, Rha J, Johnson RS, Simon MC, Keith B, and Haase VH. 2007. Hypoxia-inducible factor-2 (HIF-2) regulates hepatic erythropoietin in vivo. J Clin Invest *117*: 1068-1077.

Centanin L, Gorr TA, and Wappner P. 2010. Tracheal remodelling in response to hypoxia. Journal of insect physiology *56*: 447-454.

Wegener G. 1996. Flying insects: model systems in exercise physiology. Experientia *52*: 404-412.

L. 66: "Invertebrates have only one Hif-α paralog (Hif-1α)." I think you should finish the thought stimulated by the prior sentence about splice variants in the Hif family genes of vertebrates. This sentence might say "Invertebrates have only one Hif-1α gene, but flybase indicates four alternative transcripts in *Drosophila* and there is further evidence for splice variants in the transcriptomes of other insect species." You could revisit this in the discussion, and in so doing cover what for me is a weak point of the manuscript: is this splice variant unique to migratory locusts, orthoptera in general, or is it more broadly present in insects but heretofore overlooked? You show that in a supplementary figure but don't discuss it.

We agree with the review and revised the sentences as suggested (Line 65-67).

As you can see in our reply to the reviewer’s question 1, we have added detailed descriptions on taxonomic analysis of Hif-α transcripts in Line 108-116 and discussed the functional innovation of C-TAD-lacked Hif-1α transcripts in insects in Line 366-377.

Additionally, we have added more details in Figure 1—figure supplement 2 showing the genome duplication events of Hif-α gene in vertebrates.

L 69: "associated with altered flight performance; this association suggests the possible involvement of Hif signaling in flight adaptation (Marden et al., 2013)." I think you should cite two followup studies (https://journals.biologists.com/jeb/article/221/6/jeb171009/246/Enzyme-polymorphism-oxygen-and-injury-a-lipidomic; https://pubmed.ncbi.nlm.nih.gov/32895932/) that discuss oxidative damage, larval development, and additional aspects of flight in that species. Doing so will point readers to a broadly relevant set of studies that examine intermittent flight and ischemia reperfusion injury caused by repeatedly flying to fatigue. Making a distinction between migratory vs. intermittent fliers may stimulate important insights as work in this area progresses.

Thanks for the good suggestions, we have cited these important studies behind this sentence and given a discussion on this topic as follows:

“Additionally, oxidative damage caused by flight in insects differs in their flight physiology, behavior and life history. A sustained flight throughout life can cause a higher mortality rate to *Drosophila* (Magwere et al., 2006). Flight activity of honey bees directly leads to increased oxidative damage, which in turn detrimentally affects their flight performance and foraging ability (Margotta et al., 2018). Insects have evolved a series of adaptive strategies to cope with intermittent and migratory flight-induced oxidative stress. Glanville Fritillary butterflies carrying Sdhd M allele are associated with the activated Hif signalling, reduced metabolic rate, and larger tracheal volume in larvae, and these associations contribute to less oxidative injury in flight muscle and better flight performance during intermittent flight in adults (Marden et al., 2021; Pekny et al., 2018). Nectar feeding hawkmoths use their antioxidant stores during migratory flight and through PPP to produce an antioxidant potential to recover from oxidative damage during rest (Levin et al., 2017). While, the utilization of PPP was reported to be positively correlated with the activation of Hif pathway (Sadiku and Walmsley, 2019; Tokuda et al., 2019). Therefore, at the molecular level, the Hif pathway likely plays a central role in regulating redox homeostasis during insect flight.” (Line 321-335)

L. 76: "make migratory flights on the Tibetan Plateau". Please indicate an approximate or range of altitudes where this occurs.

Thanks, we revised the sentence to read:

“Moreover, migratory locusts can metabolically adapt to hypoxic environments and make migratory flights on the Tibetan Plateau at altitudes over 3700 meters.” (Line 79)

L. 80-90: I like this preview summary of the results, but if editors insist on a length reduction, you could save this for the Results or Conclusion sections and instead say something brief, e.g. "Here we characterize two alternative splice variants of locust Hif-1a and test the hypothesis that they perform different roles in flight physiology and migratory ability."

Thanks, we have retained this preview summary.

L. 101: "On the basis of full-length transcriptome, two transcripts of Hif-1α were observed." It is not obvious at this point or in the methods how this transcriptome was made (whole body? flight muscle?) and if/where it is accessible by others.

Thanks for the comments. The full-length transcriptome data was published in a previous study of our lab with NCBI accession number PRJNA517220. RNA samples that used for libraries construction are generated from testes of locusts. We have added a description in Line 494-498:

The full-length transcriptome was obtained from (Jiang et al., 2019). Briefly, the full-length transcripts from testis tissues of locusts were enriched by 5’-Cap capturing assay for library preparation. 200 ng of the RNA libraries were loaded on FLO-MIN106 (R9.4) flowcells and were run on a MinION or a GridION X5 (Oxford Nanopore Technologies).

L. 106: "encoded protein lacked the C-TAD". Extend this sentence by explaining what this domain does. This will inform the hypothesis that these isoforms have different functions.

Thanks, we have revised the sentence to read:

“The CDs of *Hif-1α2* were 201 bp shorter than those of *Hif-1α1* and the encoded protein lacked the C-TAD, which is required for the transitivity of Hif-1α (Figures 1C and D).” (Line 108)

L 106-109: You highlight the presence of similar splice variants in vertebrates but don't mention other insects. Your supplementary figure showing this diversity is very nice and provides an overview of the diversity of Hif-1a splice variation, along with boxes that I think show presence/absence of Hif regulatory genes in the genome (figure legend show define those boxes unambiguously). Spend a sentence or two describing what you see across insects, and perhaps in the discussion, how well you think insects have been carefully surveyed for splice variants, and what your findings may imply for insects other than locusts.

Following the reviewer’s suggestion, we added definition of these HIF regulatory proteins in the figure legend. We’d like to follow the reviewer’s suggestion and added a description of this Figure as follows:

“Evolutionary analysis revealed that such Hif-1α splice form also exists in other Orthoptera (Accession no. ON137898 and ON137899 for Deracantha onos), some birds (e.g., XP_025006307.1 for Gallus gallus and XP_013038471.1 for Anser cygnoides domesticus), and human (NP_851397.1). Additionally, the TADs of Hif-α have varied distributions among insects. In incomplete metamorphosis insects and beetles, the Hif-α protein possesses two TADs (N-TAD and C-TAD), but in flies and moths the C-TAD and its inhibitor FIH are completely missing at the genomic level. Therefore, C-TAD-lacking Hif-1α transcripts, with distinct origins, seem to commonly exist in different insect taxa (Figure 1—figure supplement 2).” (Line 108-116)

We also added a discussion on this issue in Line 321-335.

L. 263: Not clear if "In vertebrate" is a typo referring to invertebrates or is something that occurs "in vertebrates".

Sorry for the confusing description. We have revised this sentence to read:

“Vertebrate myeloid cells have shown a similar upregulation of Hif-1a protein in well-oxygenated environments.” (Line 267)

L. 274: "Therefore, Hif-1α2 was less sensitive to oxygen tension than Hif-1α1". Figure 1E indicates that Hif1a2 reaches higher levels in hypoxia, so it is arguably more sensitive to low oxygen tension (whereas your point is based on it starting from a higher baseline). To make this easier to understand, just say that "Hif1a2 had different oxygen sensitivity, being less inhibited by normoxia, and increasing to high levels in low oxygen."

We agree with the reviewer and revised this sentence as follows:

“Therefore, Hif-1α2 had different oxygen sensitivity, being less inhibited by normoxia, and increasing to high levels in low oxygen.” (Line 278)

L 335-336: "However, no such cellular hypoxia-adaptive responses were detected in the flight muscle during flight." Somewhere you should introduce the idea that the abundant tracheal system of locust flight muscle, and the relatively easy task of flying with weight support on a flight mill, may have kept the intracellular oxygen tension above the low level that triggers Hif-1a1 stability and canonical activation of hypoxia responsive genes. If you introduce that concept, it recognizes that there may be a role for Hif-1a1 in flight muscle that your experiments did not reveal, and helps form the hypothesis that Hif-1a2 has an earlier and graded activity, from a low level at normoxia and increasing as oxygen decreases, thereby protecting against oxidative damage at all levels of exertion rather than as an emergency response that may lag behind the damage.

Thanks for the insightful comments. Follow the reviewer’s suggestion, we have modified the statement in line 350 and discussed the effect of tracheal system on insect flight and compared the canonical and non-canonical role of Hif signalling in flight regulation:

“The two Hif-1α splices may coordinate their roles in long-lasting flight tasks. In locusts, the canonical role of Hif pathway is modulated by Hif-1α1, which regulates metabolic reprogramming and possibly controls tracheal growth under low oxygen tension. However, the abundant tracheal system of the locust flight muscle may keep the intracellular oxygen tension above the low level that triggers Hif-1α1 stability. Meanwhile, the relatively easy task of flying with weight support on a flight mill in the present study may render the role of Hif-1α1 in flight muscle undetectable. Nevertheless, when it comes to highly active tissue, Hif-1α1 may provide protective effects too late to prevent oxidative damage. Instead, Hif-1α2, which is expressed in normoxia and has a graded activity with decreasing oxygen, provides continuously variable expression of antioxidant genes so that protection is in place before the damage occurs. This is different from the way Hif-1α1 is typically activated only at very low oxygen tension. As shown in Figure 1—figure supplement 2, the similar transcript form of locust Hif-1α2 also exists in some other insect species and birds. Therefore, the Hif-1α2-mediated protective mechanism is possibly applicable to other flying animals, with the locust in this study as the first glimpse.” (Line 390-403)

L 355-356: "On the other hand, our study provides a potential insecticide target for locust plague control." This is presented as an afterthought. Either omit, or if you think it is an important point, begin a new paragraph and develop the argument a bit more. For instance, do you envision this as an RNAi treatment? What would be the major challenges in targeting and minimizing off-target effects, both within locusts and in other insects that one would not want to damage.

Thanks for the comments, we have omitted this statement.

Figure 5A: The oxidized DHE appears to be elevated in spaces between cells or groups of cells (myofibers). To my eye, those look like tracheae, and the red dots within fibers might be cross sections of tracheoles where they penetrate into cells. The authors may be stimulated to consider a coupled role of tracheae and muscle cells in sensing falling O2 concentration and functioning together to protect both cell types against oxidative damage. Due to their thin-walled and membranous nature, it seems possible that tracheae may be more sensitive to oxidative damage than muscle, or that muscle has high basal expression of a number of anti-oxidants but tracheae do not. The authors may wish to discuss such a scenario, and testable hypotheses that may emerge from this notion. Also, my copy has fairly low resolution images of these micrographs. The authors should make sure that their highest resolution images are published.

We appreciate the reviewer’s invaluable comments. DHE can cross intracellular membranes directly and, upon oxidation, becomes positively charged and accumulates in cells by intercalating into nucleus DNA. Muscle cells are the fusion of hundreds of myoblasts and their nuclei are pushed aside to the periphery of the myofiber below the plasma membrane. Therefore, it is more likely that in this figure the DHE signals are located in the nuclei and reflecting the ROS level of muscle cells. Here, the reviewer provided us an important information about the roles of tracheae in sensing O2 tension and protecting against oxidative damage. We have added a discussion on this issue in Line 378-389 as follows:

“Insects were the first animals capable of flying on the planet, and their respiratory system is the tracheal system. The tracheal supply to flight muscles is divided into three parts: primary tracheae, flattened tracheae and terminal tracheoles, among which the terminal tracheoles apply to the surfaces of the mitochondria and convey oxygen directly into the mitochondria (Wigglesworth and Lee, 1982). During flight, the tracheal system reaches its maximum functional requirements with little reserve capacity (Snelling et al., 2017; Snelling et al., 2012). Therefore, the tracheal volume and flight performance of insects are closely associated. In the larval stages, the tracheal system becomes oxygen-sensitive, and Hif pathway controls the growth of tracheal terminal branches toward oxygen-starved areas (Centanin et al., 2010; Henry and Harrison, 2004). The canonical role of the Hif pathway contributes to the ecological adaptation of Glanville Fritillary butterflies from genetically distinct clades (Marden et al., 2013; Marden et al., 2021; Pekny et al., 2018).”

Additionally, we have provided the high-resolution image of this figure in Figure 4A.

Figure 6 C: Competitor is misspelled.

Thank you, it has been revised.

Figure 6: Panels B: why are the these Hif genes labeled PC (which could be confused for "positive control")?Panel C: What are the SH and UB bands in panel C, and where in the figure is a shift evident?Please provide a bit more information for people like me who aren't familiar with the details of this type of gel shift assay.

Thanks for the comments. Sorry for providing the confusing description. The pcDNA here is name of vector which *Hif-1α* gene was cloned into. To avoid making ambiguous description, we have revised the pcDNA as Empty vector in Figure 6 C and we have added more details about this figure in the figure legend. (Line 973-975)

Reviewer #2 (Recommendations for the authors):Although I am not a molecular biologist, I was impressed with the comprehensiveness of the work. My only main source of confusion was the link between Hif-1alpha2, the upregulation of glucose oxidation, and its role in facilitating prolonged flight. As you state later in the paper, locust flight is only initially powered by carbohydrate catabolism, and sustained locust flight is instead powered by lipid catabolism. I could be wrong, but I suspect that upregulation of glucose oxidation may rather help to augment the power output of initial flight, when locust metabolic requirements are greatest as they must power the requirements of ascent. Thereafter, locusts switch to lipid catabolism and more prolonged flight is also facilitated by the use of thermal currents and tail winds.

We appreciate the reviewer’s invaluable comments. We agree with the reviewer that upregulation of glucose oxidation mainly helps to augment the power output of initial flight and the prolonged flight of locusts is supported by lipid oxidation, thermal currents and tail winds. And our results are in line with the reviewer’s opinion. Actually the long distance migration of locust swarms is contributed by multiple environmental and physiological factors. In the aspect of physiological level, which this study mainly focused on, a prolonged flight requires not only a sufficient energy supplement but also a maintained redox homeostasis in myocytes. Hif-1α2 regulates expression of target genes involved in glucose metabolism and anti-oxidation, and we further found that regulatory role of Hif-1α2 on anti-oxidation, but not glucose metabolism, contributes to the sustained flight of locusts. To avoid any misunderstanding on this aspect, we have revised the description in Line 21 to read:

“Hif-1α2 regulates physiological processes involved in glucose metabolism and anti-oxidation during flight and sustains flight endurance by maintaining redox homeostasis through upregulating the production of a reactive oxygen species (ROS) quencher, DJ-1.”

Reviewer #3 (Recommendations for the authors):1. Despite the relatively high levels of Hif-1a2 in resting flight muscles, the authors have shown that Hif-1a2 is further induced after the long flight (30-60min), which would elevate the levels of Hif-1a2 target genes (Figure 2D). And given that loss of Hif-1a2 does not cause any flight defects in early time points (Figure 2E), it is assumed that an increase in the level of Hif-1a2 upon long flight is critical for its function and target genes. Thus, it would have been more appropriate to perform the transcriptome analysis (Figure 3A) after the long flight to better understand transcriptional targets of elevated Hif-1a2 during the flight. Current data may not reflect complete transcriptomic changes of flight muscles after the Hif-1a2 induction and would indicate targets of Hif-1a2 in resting states. As a minimum, the 12 differentially expressed genes need to be validated before and after the long flight and in the presence and absence of Hif-1a2 to better verify the expression and function of glucose metabolism genes and DJ-1.

We appreciate the reviewer’s invaluable comments. We followed the reviewer’s suggestion and added the experiments. We tested the 11 differentially expressed genes before and after the long flight, in which locusts were forced to fly for 60 min with the presence and absence of Hif-1α2. We have replaced the Figure 4B with a new one. Meanwhile, we deleted Supplementary Figure 4 and replaced Figure 6A with a new one because of result duplication in these figures. We have revised the results to read:

“Then the glycolytic gene expression was examined when the locusts were forced to fly for 60 min after the knockdown of *Hif-1α1* and *Hif-1α2*. The expression levels of *GAPDH*, *PGX*, *GBE1*, *PGI*, *ENO*, *PFK*, *PGM*, *PYK*, *LDH* and *PDHX* were tested via qRT-PCR. The results showed that knockdown of *Hif-1α2*, rather than *Hif-1α1*, dramatically repressed flight-induced upregulation of these genes (Figure 4b).” (Line 202)

2. In the same vein, 0-hour controls for dsHif-1a1 or dsHif-1a2 are missing in Figure 4B, Supplementary Figure 4, and Figure 6A. The authors need to clarify whether Hif-1a2 activates downstream targets in resting conditions compared to the long-term flight.

We are sorry for our confusing description and prepared a new figure. Figure 4B, Supplementary Figure 4 and Figure 6A demonstrated that, these target genes were upregulated under flight conditions (15 and 60 min flight treatment), but these upregulations were repressed by the knockdown of *Hif-1α2*. 0 min here represents the resting condition. To avoid providing any misleading information again, we replaced them with new figures, i.e., Figure 4B and Figure 6A.

3. For the flight measurements in Figure 2E, the authors observed flight distance, duration, and speed up to 180 minutes for flight abilities, from which loss of Hif-1a2 showed phenotypes after 120 minutes. In Figure 4D and Figure 6E, the authors also measured the flight performance, but experiments do not seem to be identical to what is shown in Figure 2E, according to methods. The evaluation of flight performance in dsPyk or dsDJ-1 backgrounds needs to be identical to what is shown in Figure 2E as lack of energy would likely impact long flight behaviors. If it is a matter of graph formatting, the same format needs to be presented to directly compare phenotypes of different genotypes.

Thanks for the comments. We are glad to follow the reviewer’s suggestion and replaced Figure 4D and Figure 6E with newly formatted figures. We have modified the results to read:

“However, when the *PYK* in locusts was knocked down, no obvious effects were observed on flight distance, flight duration, and average flight speed in 60, 120 and 180 min tests (Figure 4D).” (Line 215-218)

“In line with Hif-1α2, the knockdown of *DJ-1* significantly repressed the flight performance of locusts in terms of flight distance (*p* = 0.0488) and average flight speed (*p* = 0.0142) in 120 min tests; and flight distance (*p* = 0.0348), flight duration (*p* = 0.0448), and average flight speed (*p* = 0.0149) in 180 min tests on flight mills (Figure 6E).” (Line 249-253)